# Pch2 orchestrates the meiotic recombination checkpoint from the cytoplasm

Esther Herruzo[1], Ana Lago-Maciel[1], Sara Baztán[1], Beatriz Santos[1,2], Jesús A. Carballo[3], Pedro A. San-Segundo[1]*

**1** Instituto de Biología Funcional y Genómica (IBFG), Consejo Superior de Investigaciones Científicas (CSIC) and University of Salamanca, Salamanca, Spain, **2** Departamento de Microbiología y Genética, University of Salamanca, Salamanca, Spain, **3** Department of Cellular and Molecular Biology. Centro de Investigaciones Biológicas Margarita Salas, Consejo Superior de Investigaciones Científicas (CSIC), Madrid, Spain

* pedross@usal.es

**Data Availability Statement:** All relevant data are within the manuscript and its Supporting Information files.

## Abstract

During meiosis, defects in critical events trigger checkpoint activation and restrict cell cycle progression. The budding yeast Pch2 AAA+ ATPase orchestrates the checkpoint response launched by synapsis deficiency; deletion of *PCH2* or mutation of the ATPase catalytic sites suppress the meiotic block of the *zip1Δ* mutant lacking the central region of the synaptonemal complex. Pch2 action enables adequate levels of phosphorylation of the Hop1 axial component at threonine 318, which in turn promotes activation of the Mek1 effector kinase and the ensuing checkpoint response. In *zip1Δ* chromosomes, Pch2 is exclusively associated to the rDNA region, but this nucleolar fraction is not required for checkpoint activation, implying that another yet uncharacterized Pch2 population must be responsible for this function. Here, we have artificially redirected Pch2 to different subcellular compartments by adding ectopic Nuclear Export (NES) or Nuclear Localization (NLS) sequences, or by trapping Pch2 in an immobile extranuclear domain, and we have evaluated the effect on Hop1 chromosomal distribution and checkpoint activity. We have also deciphered the spatial and functional impact of Pch2 regulators including Orc1, Dot1 and Nup2. We conclude that the cytoplasmic pool of Pch2 is sufficient to support the meiotic recombination checkpoint involving the subsequent Hop1-Mek1 activation on chromosomes, whereas the nuclear accumulation of Pch2 has pathological consequences. We propose that cytoplasmic Pch2 provokes a conformational change in Hop1 that poises it for its chromosomal incorporation and phosphorylation. Our discoveries shed light into the intricate regulatory network controlling the accurate balance of Pch2 distribution among different cellular compartments, which is essential for proper meiotic outcomes.

## Author summary

During gametogenesis, the number of chromosomes is reduced by half and it returns to the normal ploidy when the two gametes fuse during fertilization. Meiosis lies at the heart of gametogenesis because it is the specialized cell division making possible the reduction in ploidy. The fidelity in this process is essential to maintain the chromosome

**Funding:** This work was supported by the grant RTI2018-099055-B-I00 from Ministry of Science, Innovation and Universities (MCIU/AEI/FEDER, EU) of Spain to PSS and JAC. EH was partially supported by the grant CSI259P20 from the "Junta de Castilla y León". The IBFG is supported in part by an institutional grant from the "Junta de Castilla y León, Ref. CLU-2017-03 co-funded by the P.O. FEDER de Castilla y León 14-20". The funders had no role in study design, data collection and analysis, decision to publish, or preparation of the manuscript.

**Competing interests:** The authors have declared that no competing interests exist.

complement characteristic of the species and to avoid aneuploidies. Meiotic cells possess an intricate surveillance network that monitors crucial meiotic events. In response to defects in synapsis and recombination, the meiotic recombination checkpoint blocks meiotic cell cycle progression, thus avoiding aberrant chromosome segregation and formation of defective gametes. The AAA+ ATPase Pch2 is an essential component of the checkpoint response triggered by the recombination defects occurring in the *zip1Δ* mutant lacking the central region of the synaptonemal complex. Pch2 supports proper chromosomal localization and phosphorylation of the Hop1 axial component required for the ensuing checkpoint response. We reveal here the biological relevance of a cytoplasmic population of Pch2 that is necessary for meiotic events occurring on chromosomes. Using a variety of strategies, we demonstrate that the checkpoint activating function of Pch2 takes place outside the nucleus, whereas the nuclear accumulation of Pch2 has deleterious consequences. Our work highlights the importance of nucleocytoplasmic communication for a balanced distribution of Pch2 among different subcellular compartments and how it impinges on Hop1 dynamics, which is crucial for proper completion of the meiotic program.

## Introduction

Sexually-reproducing organisms conduct a specialized type of cell division called meiosis. During this process, chromosome ploidy is reduced by half, due to two rounds of nuclear divisions preceded by only one round of DNA replication. Meiosis is characterized by its long prophase I stage, where the following highly regulated processes take place: pairing, synapsis and recombination. Recombination initiates with the introduction of programmed DNA double-strand breaks (DSBs) catalyzed by Spo11 and its associated proteins [1]. These breaks are then processed and repaired, part of them as crossovers (CO) [2], to establish physical connections between homologous chromosomes essential to direct their proper segregation [3]. As recombination proceeds, chromosome synapsis occurs by the polymerization of the synaptonemal complex (SC) connecting the axes of paired homologs. This conserved highly-organized proteinaceous structure provides the adequate environment for properly regulated recombination [4]. The SC comprises a central region, which in budding yeast is mainly composed by the transverse filament Zip1 protein [5] including also the so-called central element formed by Ecm11 and Gcm2 [6], and two lateral elements (LEs) made of Hop1, Red1 and Rec8 [7–11].

Progression and completion of these complex meiotic events are carefully monitored by a surveillance mechanism, called the meiotic recombination checkpoint, that triggers cell-cycle arrest in response to defective synapsis and/or recombination thus preventing meiotic chromosome missegregation [12]. Over the years, several components in this pathway have been identified using *S. cerevisiae* mutants defective in different meiotic events (i.e., *zip1Δ* or *dmc1Δ*) as genetic tools to activate the checkpoint. Current evidence indicates that the unique signal leading to checkpoint activation is the presence of unrepaired DSBs, and argues against the existence of a synapsis checkpoint in yeast [13]. Moreover, a unified logic for the checkpoint response triggered both in the DSB repair-deficient *dmc1Δ* mutant and in the synapsis-defective *zip1Δ* mutant has been recently proposed [14]. In these mutants, unrepaired resected meiotic DSBs recruit the Mec1^ATR-Ddc2^ATRIP kinase sensor complex [15], which is responsible for Hop1^HORMAD1,2 phosphorylation at various consensus S/T-Q sites. In particular, Red1-mediated Hop1 phosphorylation at T318 is required to recruit Mek1^CHK2 to chromosome axes [16–18], thus favoring its dimerization and *trans* autophosphorylation events required for full Mek1 activation [19,20]. In turn, activated Mek1 stabilizes Hop1-T318

phosphorylation in a positive feed-back loop [17]. In *zip1Δ*, the phosphorylation status of Hop1-T318 is critically modulated by the AAA+ ATPase Pch2[TRIP13], which is also responsible for Hop1 chromosomal abundance and dynamics [21] (see below). Once Mek1 is fully activated, it inhibits DSB repair by intersister recombination in part by preventing Rad54-Rad51 complex formation via direct phosphorylation of Rad54 and Hed1 [22,23]. On the other hand, active Mek1 also blocks meiotic cell cycle progression by direct inhibition of Ndt80 [24], a transcription factor driving the expression of genes encoding proteins required for prophase I exit, such as the polo-like kinase Cdc5 and the type-B Clb1 cyclin [25–27]. A negative feedback loop has been described in which active Ndt80 downregulates Mek1 activity through Cdc5-dependent Red1 degradation [28]. Checkpoint-induced Mek1 activation also leads to high levels of the Swe1 kinase, which inhibits Cdc28[CDK1] by phosphorylation at Tyr19, further contributing to slow down meiotic progression [29,30].

Pch2 is an evolutionarily conserved AAA+ ATPase initially discovered in *S. cerevisiae* [31], but also present in other organisms that undergo synaptic meiosis such as worms, fruit flies, plants and mammals. Budding yeast Pch2 is meiosis specific and it has been implicated in a vast number of meiotic processes. The most thoroughly characterized role of Pch2 (known as TRIP13 in mammals) is the action on proteins that share a peptide-binding domain termed the HORMA domain (for Hop1, Rev7 and MAD2) [32]. Since Hop1 is required for meiotic DSB formation, during wild-type meiosis Pch2[TRIP13] excludes Hop1[HORMAD1,2] from fully synapsed meiotic chromosomes, constituting a feedback mechanism suppressing further recombination in regions that have already synapsed [33–36]. Nevertheless, there are particular genomic regions that retain Hop1 and undergo continuous breakage despite normal SC deposition [37]. On the contrary, in the synapsis-defective *zip1Δ* mutant, Pch2 is critically required for the meiotic recombination checkpoint, promoting proper loading of Hop1 on unsynapsed chromosome axes and supporting sufficient levels of Hop1-T318 phosphorylation driving the downstream checkpoint response [21]. In addition, in *C. elegans* and mammals, PCH-2[TRIP13] also modulates the conformational state of MAD2 to accomplish a satisfactory spindle assembly checkpoint response [38–40]. As an AAA+ ATPase, Pch2 assembles into homo-hexamers with a central pore loop; this structure is critical for producing conformational changes on HORMA-containing proteins via cycles of nucleotide binding and hydrolysis [41,42]. HORMAD proteins can assemble in multiprotein complexes through the binding of the HORMA domain core to the so-called closure motif in interacting partners. Both, the Hop1 binding partner Red1, and Hop1 itself, contain closure motifs that direct Hop1 assembly on chromosome axes [43,44]. Furthermore, structural studies *in vitro* indicate that Hop1 can adopt two stable conformations in solution [44], similar to what is described for HORMAD proteins of higher eukaryotes [40,45,46]. In the self-closed state of Hop1, the C-terminal closure motif bound to the HORMA domain is wrapped by the safety belt region located at the C-terminal part of the HORMA domain core, locking the closure motif. In the more extended conformation called 'unbuckled', the safety belt is disengaged allowing the binding to a new closure motif. It has been proposed that Pch2 catalyzes the transition from the "closed" to the "unbuckled" conformation releasing the safety belt lock from the HORMA domain core [44]. Contrary to what is described in plants, worms and mammals, this action of Pch2 does not involve the p31[COMET] co-factor, which is absent in budding yeast [38,47–49].

Studies of budding yeast Pch2 localization on spread chromosomes have revealed that it mainly localizes to the rDNA region, but some foci are also detected on fully synapsed meiotic chromosomes [31,50]. This localization pattern differs in a synapsis-deficient situation that activates the checkpoint (i.e., *zip1Δ* mutant), where Pch2 loses its association to chromosomes and it is only concentrated on the nucleolar region [21,31]. The presence of Pch2 in the rDNA, promoted by its interaction with Orc1, is required to exclude Hop1 from this region

preventing potentially harmful DSB formation in this highly repetitive genomic location [31,51,52]. Additional factors that regulate Pch2 distribution between the chromosomes and the rDNA are chromatin modifiers, such as the Dot1 histone methyltransferase and the Sir2 histone deacetylase [20,31,51,53,54], the Nup2 nucleoporin [37], and the Top2 toposisomerase [55]. We have recently demonstrated that Pch2 also shows a cytoplasmic localization, and that the Orc1-dependent nucleolar population of Pch2 is actually dispensable for the meiotic recombination checkpoint [56], leaving the prevention of DSB formation at the rDNA as the sole known role of nucleolar Pch2.

Here, we reveal where the population of Pch2 that is relevant for the *zip1Δ*-induced meiotic recombination checkpoint localizes in the cell. We show that in the *zip1Δ orc1-3mAID* mutant, Pch2 is exclusively detected in the cytoplasm and the checkpoint is fully active, strongly suggesting that Pch2 promotes Hop1 association to unsynapsed meiotic chromosomes from the cytoplasm. Our analyses of the meiotic outcomes resulting from artificially forced nuclear import or export of Pch2, and from its sequestration at the inner face of the plasma membrane, further support the notion that the role of Pch2 in checkpoint activation is executed from outside the nucleus. We have also investigated the contribution of other proteins that control Pch2 localization. We demonstrate that when Pch2 is absent from the rDNA, Dot1 is no longer required for the checkpoint, indicating that the unique role of Dot1 in the meiotic recombination checkpoint is to maintain Pch2 nucleolar confinement to avoid harmful Pch2 accumulation on unsynapsed chromosomes. We additionally show that Nup2 also participates in Pch2 subcellular distribution; in the absence of *NUP2*, an increased cytoplasmic accumulation of Pch2 occurs and, consistently, the *zip1Δ*-induced checkpoint remains intact. In conclusion, we show for the first time the existence of a functionally relevant cytoplasmic pool of Pch2 in meiotic yeast cells and we define key requirements for a precise balance of Pch2 distribution among different subcellular compartments critical for successful meiotic function.

## Results

### Pch2 localizes to the cytoplasm in the absence of Orc1, but the *zip1Δ*-induced checkpoint remains active

Using chromosome spreading and a conditional auxin-inducible *orc1-3mAID* degron allele, we have previously described that Pch2 is not recruited to the nucleolus (rDNA) in the absence of Orc1; however, the *zip1Δ*-triggered meiotic recombination checkpoint remains fully functional in this situation, demonstrating that Pch2 nucleolar localization is dispensable for the checkpoint. Furthermore, in the *zip1Δ* mutant lacking Orc1, Pch2 is not detected whatsoever associated to meiotic chromosomes, but the meiotic checkpoint is still functional [56]. This observation raises the possibility of a chromosome-independent fraction of Pch2 that may sustain the checkpoint response. To elucidate where the Pch2 population that is relevant for checkpoint function is present in the cell, we studied Pch2 localization in whole meiotic cells in different conditions. To this end, we used the $P_{HOP1}$-GFP-PCH2 construct previously described [56], which expresses *GFP-PCH2* from the meiosis-specific *HOP1* promoter [10]. We have already shown that expression of *GFP-PCH2* from its own *PCH2* promoter results in very low levels of the tagged protein; however, the $P_{HOP1}$-GFP-PCH2 construct produces GFP-tagged Pch2 at near physiological levels, comparable to those of the native untagged protein [56]. Thus, all the *GFP-PCH2* variants employed throughout this paper are driven by the *HOP1* promoter, but for simplicity this feature is omitted from the relevant genotypes shown in the text and figures.

To examine Pch2 subcellular distribution in *zip1Δ orc1-3mAID* whole meiotic cells we integrated the *GFP-PCH2* construct into the genome at the *PCH2* locus. We first checked that the

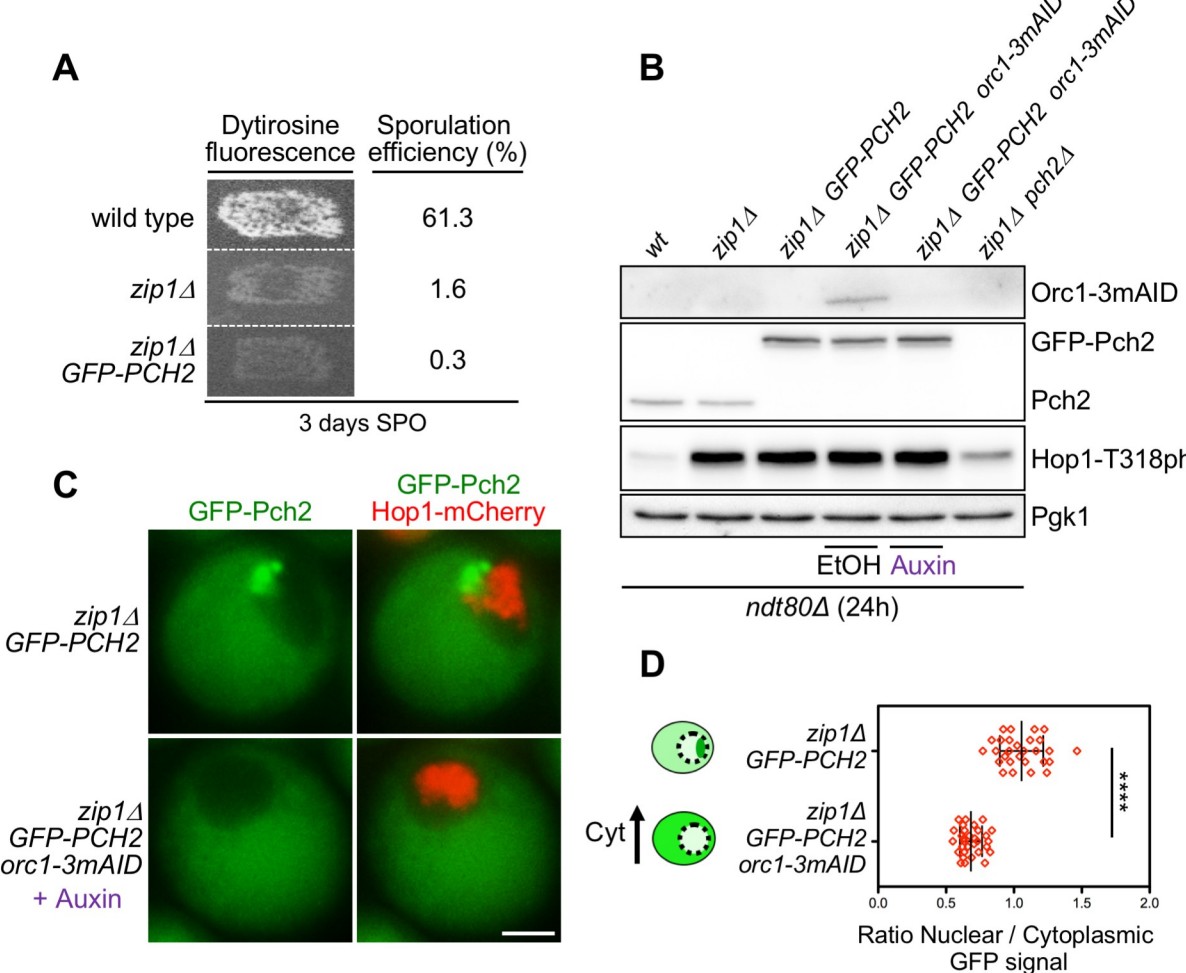

**Fig 1. Cytoplasmic accumulation of Pch2 upon Orc1 depletion supports checkpoint activity. (A)** Functional analysis of the GFP-tagged version of *PCH2*. Dityrosine fluorescence, as a visual indicator of sporulation, and sporulation efficiency were examined after 3 days on sporulation plates. Strains are DP421 (wild type), DP422 (*zip1Δ*) and DP1621 (*zip1Δ GFP-PCH2*). **(B)** Western blot analysis of Orc1-3mAID production (detected with an anti-mAID antibody), GFP-Pch2 and Pch2 production (detected with an anti-Pch2 antibody), and Hop1-T318 phosphorylation. Pgk1 was used as loading control. Strains are DP424 (*ndt80Δ*), DP428 (*ndt80Δ zip1Δ*), DP1640 (*ndt80Δ zip1Δ GFP-PCH2*), DP1630 (*ndt80Δ zip1Δ orc1-3mAID GFP-PCH2*) and DP881 (*ndt80Δ zip1Δ pch2Δ*). EtOH or auxin (500 μM) was added to *orc1-3mAID* cultures at 12 h. Samples were collected at 24 h after meiotic induction. **(C)** Fluorescence microscopy analysis of GFP-Pch2 (green) and Hop1-mCherry (red) distribution in whole meiotic cells 16 h after meiotic induction. Representative cells are shown. Scale bar, 2 μm. **(D)** Quantification of the ratio between the nuclear (including the nucleolar) and cytoplasmic mean GFP fluorescent signal. Error bars: SD. The cartoon illustrates the subcellular localization of GFP-Pch2 (green) in the different conditions. The strains in (C) and (D) are DP1636 (*zip1Δ GFP-PCH2*) and DP1633 (*zip1Δ orc1-3mAID GFP-PCH2*). Auxin (500 μM) was added to the *orc1-3mAID* culture 12 hours after meiotic induction.

GFP-Pch2 protein is completely functional, as evidenced by the tight sporulation block of the *zip1Δ GFP-PCH2* strain, similar to that of *zip1Δ* (Fig 1A). Consistent with our previous results using strains harboring untagged or 3HA-tagged Pch2 [56], we confirmed that the checkpoint remains fully functional in the absence of Orc1; that is, in the *zip1Δ orc1-3mAID GFP-PCH2* strain treated with auxin. Checkpoint proficiency was manifested, in a prophase-arrested *ndt80Δ* background, by high levels of Hop1-T318 phosphorylation when Orc1 is depleted, also comparable to those of *zip1Δ* (Fig 1B). Next, we analyzed GFP-Pch2 and Hop1-mCherry

subcellular distribution by fluorescence microscopy in live meiotic cells. Since Hop1-mCherry does not fully support checkpoint function, all the strains used in this work harboring *HOP1-mCherry* were heterozygous for this construct (*HOP1-mCherry/HOP1*). Using chromosome spreading we confirmed that in these heterozygous strains the Hop1 protein normally decorates chromosome axes and is excluded from the rDNA region (S1 Fig). Furthermore, strains harboring *HOP1-mCherry* were only used for localization and staging purposes, not for functional analyses. In the *zip1Δ* mutant, GFP-Pch2 localized to a discrete region at one side of the nucleus that does not overlap with Hop1-mCherry. According with the well-characterized Pch2 localization on *zip1Δ* chromosome spreads [21,31,56], this discrete region corresponds to the nucleolus. In addition, GFP-Pch2 was also detected in the cytoplasm, displaying a diffuse homogenous signal (Fig 1C). In contrast, and consistent with the lack of Pch2 nucleolar localization upon Orc1 depletion observed by immunofluorescence of chromosome spreads [51,56], GFP-Pch2 exclusively localized to the cytoplasm in *zip1Δ orc1-3mAID* cells (Fig 1C). Quantification of the ratio between nuclear (including nucleolus) and cytoplasmic GFP signal confirmed the cytoplasmic accumulation of Pch2 in the absence of Orc1 (Fig 1D). Importantly, despite the altered subcellular distribution, total GFP-Pch2 protein levels were unaltered when Orc1 was depleted (Fig 1B). Since the checkpoint remains completely active in the *zip1Δ orc1-3mAID* mutant, as evidenced by the high levels of Hop1-T318 phosphorylation (Fig 1B), and checkpoint activity still depends on Pch2 [56], these results suggest that the cytoplasmic population of Pch2 is proficient to promote Hop1-Mek1 activation.

## Redirecting Pch2 subcellular distribution

To further analyze how Pch2 subcellular distribution impacts on checkpoint function we fused a Nuclear Export Signal (NES) or a Nuclear Localization Signal (NLS) to Pch2 in order to force its localization outside or inside the nucleus, respectively. Canonical NES and NLS sequences were inserted between the *GFP* and *PCH2* coding sequences in centromeric plasmids containing the $P_{HOP1}$-*GFP-PCH2* construct (S2A Fig; see Materials and Methods for details). These plasmids were transformed into *zip1Δ* strains also harboring *HOP1-mCherry* as a marker both for the nucleus and for meiotic prophase stage. Live meiotic cells were analyzed by fluorescence microscopy to examine Pch2 localization. We found that, unlike the wild-type GFP-Pch2 protein, the GFP-NES-Pch2 version was largely excluded from the nucleolus and was almost exclusively present in the cytoplasm (S2A and S2B Fig). In contrast, GFP-NLS-Pch2 strongly accumulated in the nucleolus and also showed a diffuse pan-nuclear signal (S2A and S2B Fig). Thus, these constructs are useful tools to explore the effect of biased Pch2 subcellular localization.

To avoid issues derived from plasmid-loss events and from the inherent variability in plasmid copy number among individual cells in the culture, we generated strains in which the *GFP-PCH2* construct, as well as the *GFP-NES-PCH2* and *GFP-NLS-PCH2* derivatives, were integrated into the genome at the *PCH2* locus. We generated both homozygous (*GFP-PCH2/ GFP-PCH2*) and hemizygous (*GFP-PCH2/pch2Δ*) versions of these diploid strains; the levels of GFP-Pch2 in the hemizygous strains (*hem*) were comparable to those of the endogenous Pch2, whereas the homozygous (*hom*) strains showed increased GFP-Pch2 amount (Fig 2A). We used these different variants to explore the impact of forced Pch2 localization on sporulation efficiency both in unperturbed meiosis and in checkpoint-inducing conditions; that is, in *ZIP1* and *zip1Δ* backgrounds, respectively. We found that, in *ZIP1* background, all GFP-Pch2, GFP-NES-Pch2 and GFP-NLS-Pch2 versions, when expressed either in homozygous or hemizygous strains, supported sporulation to the same levels as the wild type harboring untagged *PCH2* or the *pch2Δ* mutant did (Fig 2B and 2C; light grey bars). When we analyzed the impact

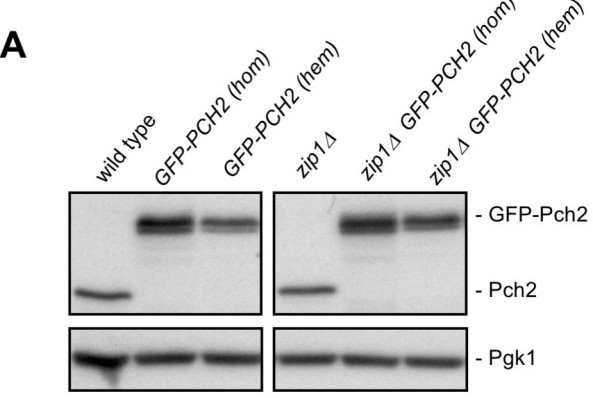

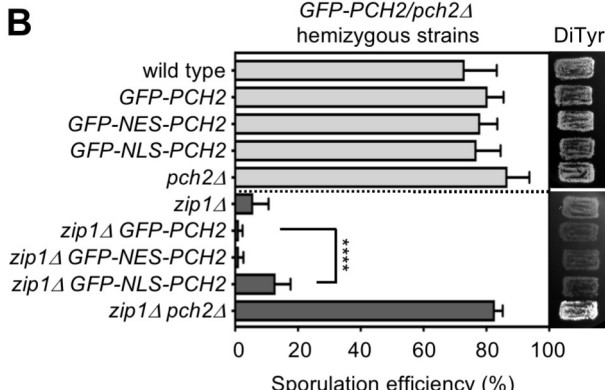

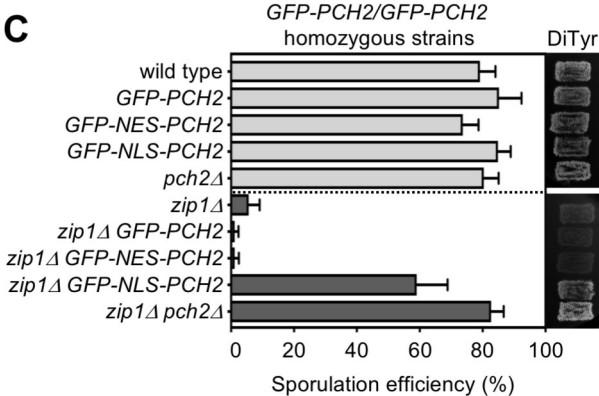

**Fig 2. Analysis of diploid strains harboring the different versions of *GFP-PCH2* integrated at its genomic locus in hemizygosis and homozygosis. (A)** Western blot analysis of GFP-Pch2 and Pch2, detected with an anti-Pch2 antibody. Pgk1 was used as loading control. Hemizygous strains are *GFP-PCH2/pch2Δ*. Strains are DP421 (wild type), DP1620 (*GFP-PCH2[hom]*), DP1624 (*GFP-PCH2[hem]*), DP422 (*zip1Δ*), DP1621 (*zip1Δ GFP-PCH2[hom]*) and DP1625 (*zip1Δ GFP-PCH2[hem]*). **(B, C)** Sporulation efficiency, assessed by microscopic counting of asci, and dityrosine fluorescence (DiTyr), as a visual indicator of sporulation, were examined after 3 days on sporulation plates. Error bars, SD; n = 3. At least 300 cells were counted for each strain. Light grey and dark grey bars correspond to *ZIP1* and *zip1Δ* strains, respectively. Strains in (B) are: DP421 (wild type), DP1624 (*GFP-PCH2*), DP1685 (*GFP-NES-PCH2*), DP1699 (*GFP-NLS-PCH2*), DP1023 (*pch2Δ*), DP422 (*zip1Δ*), DP1625 (*zip1Δ GFP-PCH2*), DP1686 (*zip1Δ GFP-NES-PCH2*), DP1701 (*zip1Δ GFP-NLS-PCH2*), and DP1029 (*zip1Δ pch2Δ*). Strains in (C) are: DP421 (wild type), DP1620 (*GFP-PCH2*), DP1669 (*GFP-NES-PCH2*), DP1695 (*GFP-NLS-PCH2*), DP1023 (*pch2Δ*), DP422 (*zip1Δ*), DP1621(*zip1Δ GFP-PCH2*), DP1670 (*zip1Δ GFP-NES-PCH2*), DP1696 (*zip1Δ GFP-NLS-PCH2*), and DP1029 (*zip1Δ pch2Δ*).

of these *PCH2* variants on *zip1Δ* strains, we found that sporulation was blocked in *zip1Δ GFP-PCH2* and *zip1Δ GFP-NES-PCH2*, both in the homozygous and hemizygous versions (Fig 2B and 2C; dark grey bars). In contrast, the sporulation block was either slightly or largely released in the hemizygous or homozygous *zip1Δ GFP-NLS-PCH2* strains, respectively (Fig 2B and 2C; dark grey bars). These initial observations suggest that GFP-NES-Pch2 is checkpoint proficient, whereas increased dosage of GFP-NLS-Pch2 compromises checkpoint function. We decided to use the hemizygous versions of all GFP-tagged *PCH2* constructs producing near physiological protein levels, as well as the homozygous *GFP-NLS-PCH2* for further comprehensive analyses of the functional impact of Pch2 localization.

We first thoroughly analyzed the subcellular localization of the different genomically-expressed GFP-tagged Pch2 versions in both *ZIP1* and *zip1Δ* live meiotic prophase I cells (Figs 3 and S3). Consistent with previous results using chromosome spreading, in *ZIP1* cells, the wild-type GFP-Pch2 was concentrated in a distinctive region inside the nucleus lacking Hop1-mCherry signal that corresponds to the nucleolus and, also, was detected in fainter discrete chromosomal foci. In addition, GFP-Pch2 displayed a diffuse cytoplasmic signal (Fig 3Aa). In the *zip1Δ* mutant, GFP-Pch2 was lost from the chromosomes and was only found in the nucleolus and cytoplasm (Fig 3Ab). In contrast, GFP-NES-Pch2 was mostly present in the cytoplasm in *ZIP1* and *zip1Δ* cells (Fig 3Ac-3Ad); only a very weak nuclear signal remained in some cells, especially in the *ZIP1* strain (Figs 3Ac and S3), that likely corresponds to remnants of the nucleolar protein. In fact, quantification of the ratio between the nuclear (including nucleolus) and cytoplasmic GFP fluorescence revealed a significant reduction in this ratio in *GFP-NES-PCH2* strains compared to *GFP-PCH2* (Fig 3B). On the other hand, GFP-NLS-Pch2 was heavily accumulated in the nucleolus (Fig 3Ae-3Af) and, in the case of *ZIP1* cells also in putative chromosomal foci (Fig 3Ae); the nuclear/nucleolar accumulation was even more conspicuous in homozygous *GFP-NLS-PCH2* strains (Fig 3Ag-j and 3B). Interestingly, GFP-NLS-Pch2 was also diffusely localized in the nucleoplasm, particularly in homozygous *GFP-NLS-PCH2* cells (Figs 3Ai-3Aj and S3). Thus, the integrated GFP-NES-Pch2 variant drives the cytoplasmic accumulation of the protein, whereas GFP-NLS-Pch2 forces its localization inside the nucleus, predominantly, but not only, in the nucleolus.

We also quantified the Hop1-mCherry nuclear signal in all these situations with altered Pch2 localization (Fig 3C). In *ZIP1* strains, Hop1-mCherry levels were slightly increased in *GFP-NES-PCH2* cells, consistent with the notion that Pch2 was largely excluded from the nucleus/nucleolus and Hop1 eviction from synapsed chromosomes and the rDNA region would be impaired, as described [21,31,57]. Consequently, NLS-driven nuclear accumulation of Pch2 resulted in reduced Hop1-mCherry levels both in *ZIP1* and *zip1Δ* strains. Curiously, the amount of nuclear Hop1-mCherry was also somewhat reduced in *zip1Δ GFP-NES-PCH2* cells despite the fact that Pch2 is not normally associated to unsynapsed chromosomes.

## Differential effect of altered Pch2 subcellular distribution on Hop1 localization in synapsed versus unsynapsed chromosomes

To obtain more detailed information, we also analyzed the localization of GFP-Pch2, GFP-NES-Pch2 and GFP-NLS-Pch2, together with that of Hop1, on pachytene chromosome spreads from prophase-arrested *ndt80Δ* cells in both *ZIP1* and *zip1Δ* backgrounds. For clarity, we first describe the localization patterns in *ZIP1* cells. Consistent with the known localization of Pch2 and the observations in live meiotic cells, the wild-type GFP-Pch2 protein localized mainly to the nucleolus (Fig 4Aa). As previously reported, the SC-associated Pch2 protein was barely detectable with this technique in the BR strain background [21,56] and only faint chromosomal GFP-Pch2 foci could be occasionally observed upon image overexposure (S4 Fig).

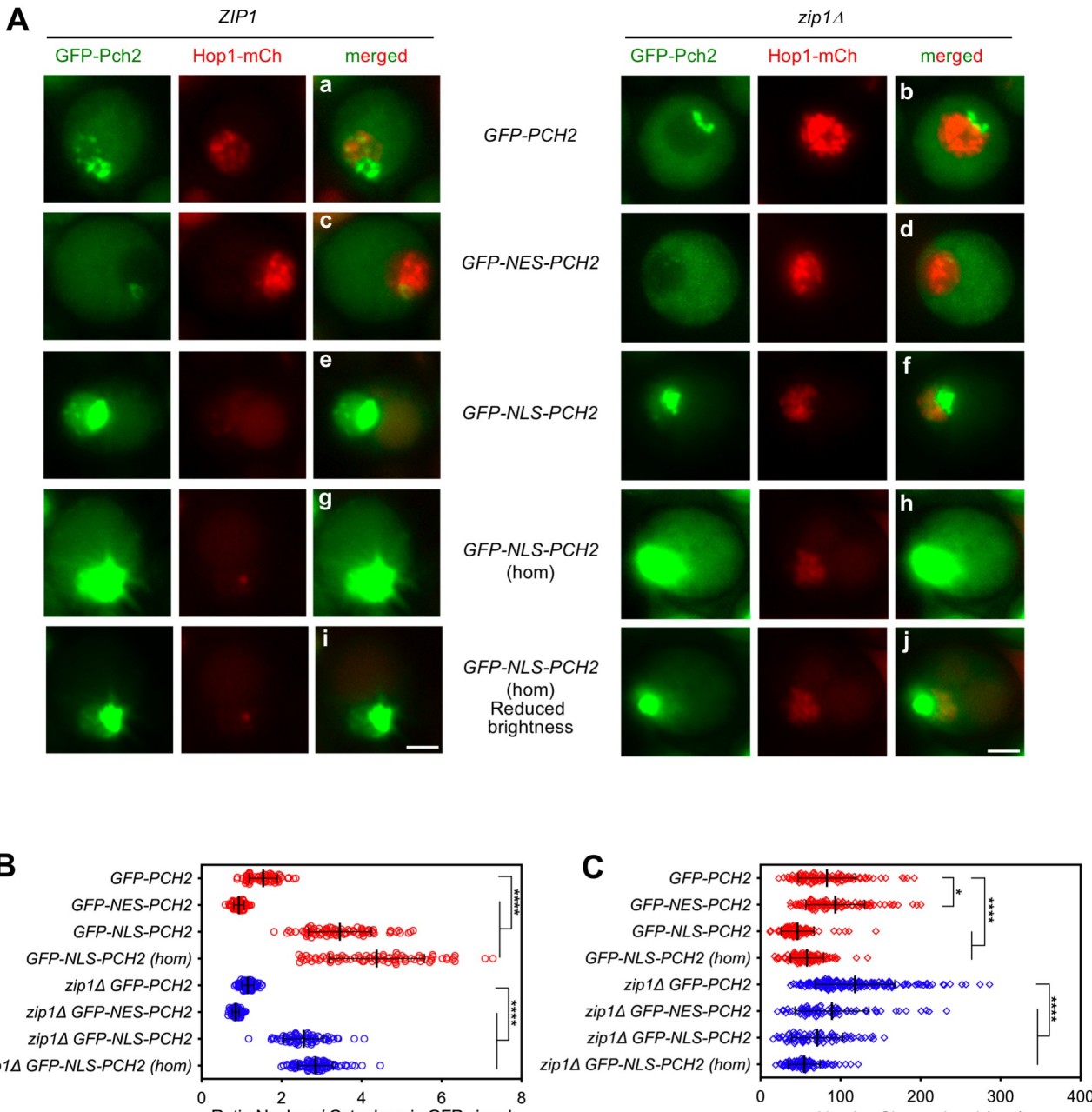

**Fig 3. Subcellular localization of GFP-NES-Pch2 and GFP-NLS-Pch2. (A)** Fluorescence microscopy analysis of genomically-expressed GFP-Pch2, GFP-NES-Pch2 or GFP-NLS-Pch2 (green) and Hop1-mCherry (red) localization in *ZIP1* and *zip1Δ* whole meiotic cells 16 h after meiotic induction. Representative individual cells are shown. Representative fields are shown in S3 Fig. All images were acquired and processed with similar settings, but in the case of *GFP-NLS-PCH2* homozygous *(hom)* cells, additional panels (i, j) with reduced GFP brightness are presented for better visualization of the strong original signal. Scale bar, 2 μm. **(B)** Quantification of the ratio of nuclear (including nucleolar) to cytoplasmic mean GFP fluorescent signal. Error bars: SD. **(C)** Quantification of the Hop1-mCherry fluorescent signal. Error bars: SD; a.u., arbitrary units. In (B) and (C), red and blue symbols correspond to *ZIP1* and *zip1Δ* strains, respectively. Strains are: DP1650 (*GFP-PCH2*), DP1687 (*GFP-NES-PCH2*), DP1700 (*GFP-NLS-PCH2*), DP1697 (*GFP-NLS-PCH2[hom]*), DP1651 (*zip1Δ GFP-PCH2*), DP1688 (*zip1Δ GFP-NES-PCH2*), DP1702 (*zip1Δ GFP-NLS-PCH2*), and DP1698 (*zip1Δ GFP-NLS-PCH2[hom]*).

Nevertheless, in agreement with a normal distribution of Pch2 in synapsed *GFP-PCH2* nuclei, Hop1 displayed its characteristic weak and discontinuous signal, and was excluded from the nucleolus (Fig 4Aa). In *GFP-NES-PCH2* nuclei, Pch2 association to chromatin was largely lost

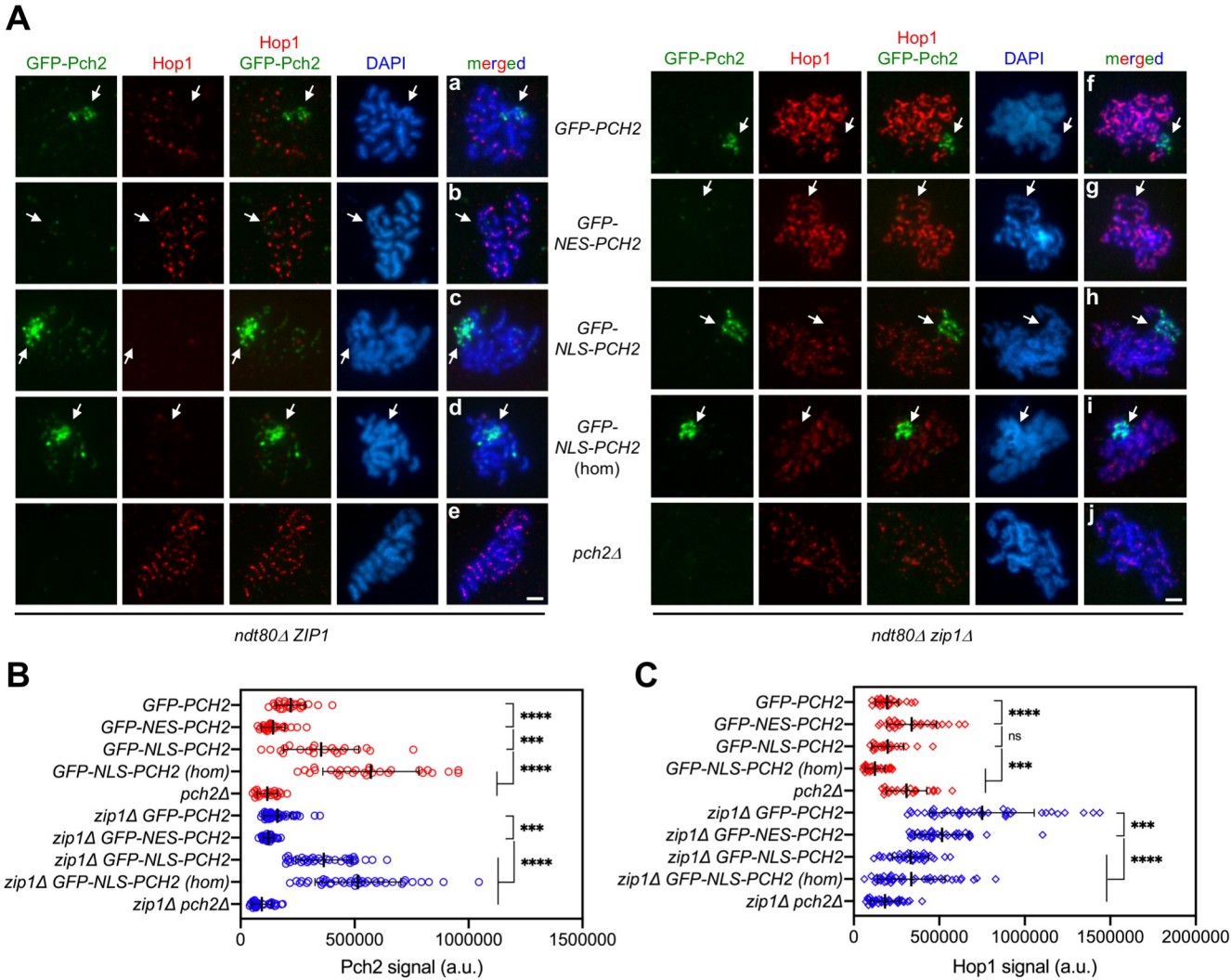

**Fig 4. Impact of GFP-NES-Pch2 and GFP-NLS-Pch2 on the pattern of Hop1 chromosomal localization. (A)** Immunofluorescence of meiotic chromosomes stained with anti-GFP antibodies (to detect GFP-Pch2; green), anti-Hop1 antibodies (red) and DAPI (blue). Representative *ZIP1* and *zip1Δ* nuclei, as indicated, are shown. Arrows point to the rDNA region. Spreads were prepared from *ndt80Δ* strains at 24 h. Scale bar, 2 μm. **(B, C)** Quantification of the GFP-Pch2 and Hop1 signal, respectively. Error bars: SD; a.u., arbitrary units. Red and blue symbols correspond to *ZIP1* and *zip1Δ* strains, respectively. Strains are: DP1654 (*GFP-PCH2*), DP1725 (*GFP-NES-PCH2*), DP1729 (*GFP-NLS-PCH2*), DP1768 (*GFP-NLS-PCH2 [hom]*), DP1058 (*pch2Δ*), DP1655 (*zip1Δ GFP-PCH2*), DP1726 (*zip1Δ GFP-NES-PCH2*), DP1730 (*zip1Δ GFP-NLS-PCH2*), DP1769 (*zip1Δ GFP-NLS-PCH2 [hom]*) and DP881 (*zip1Δ pch2Δ*).

and, accordingly, Hop1 displayed a more intense signal also moderately covering the rDNA region (Fig 4Ab, 4B and 4C), resembling the situation in the *pch2Δ* mutant (Fig 4Ae and 4C). In contrast, Pch2 densely decorated the nucleolar region in *GFP-NLS-PCH2* nuclei (Fig 4Ac) and the chromosomal foci were more visible, especially in the homozygous *GFP-NLS-PCH2* strain (Fig 4Ad). The increased presence of chromosome-associated GFP-NLS-Pch2 correlated with decreased abundance of Hop1 on spread nuclei (Fig 4B and 4C). Thus, in the context of synapsed chromosomes, the reduction of nuclear Pch2 (*GFP-NES-PCH2*) leads to increased Hop1 localization, including also the rDNA, like the complete lack of Pch2 (*pch2Δ*) does, whereas the nuclear accumulation of Pch2 (*GFP-NLS-PCH2*) counteracts Hop1 chromosomal localization.

We next describe the localization patterns of Hop1 and the different variants of GFP-tagged Pch2 in *zip1Δ* nuclei; that is, in a checkpoint-inducing condition. In *zip1Δ GFP-PCH2* nuclei, Pch2 was exclusively present in the nucleolus, and Hop1 showed the typical linear continuous pattern along unsynapsed axes, being excluded from the rDNA region (Fig 4Af). In turn, Pch2 was no longer present in the nucleolus in *zip1Δ GFP-NES-PCH2* nuclei, but Hop1 remained quite continuous along the axes also including the rDNA (Fig 4Ag). Thus, unlike the *zip1Δ pch2Δ* double mutant that displays discontinuous Hop1 localization even in the prophase-arrested *ndt80Δ* background (Fig 4Aj; [21]), the broad chromosomal (non rDNA) distribution of Hop1 in *zip1Δ* is not largely altered when Pch2 is forced out of the nucleus in *zip1Δ GFP-NES-PCH2*, arguing that, in *zip1Δ* nuclei, Pch2 is capable of promoting proper axial Hop1 localization from its cytoplasmic location. However, although the pattern of Hop1 localization was not significantly altered in *zip1Δ GFP-NES-PCH2*, the intensity of Hop1 signal was reduced (Fig 4C), suggesting that Hop1 loading and/or turnover on the axes is compromised, perhaps due to residual Pch2 present in the nucleus during its transit towards the cytoplasm. On the other hand, *zip1Δ GFP-NLS-PCH2* nuclei exhibited a marked accumulation of Pch2 in the nucleolus (Fig 4Ah), but no Pch2 association with unsynapsed axes was detected even with higher Pch2 dosage (*zip1Δ GFP-NLS-PCH2* homozygous strain; Fig 4Ai). Despite that, Hop1 axial linearity and quantity was drastically diminished (Fig 4Ah-i and 4C) suggesting that the increased abundance of Pch2 inside the nucleus, specifically in the nucleoplasm (Figs 3Ai-3Aj and S3), drives Hop1 chromosomal removal and/or that the depletion of cytoplasmic Pch2 impairs Hop1 binding to chromosome axes.

## The predominantly cytoplasmic GFP-NES-Pch2 version supports checkpoint activity

To determine how Pch2 localization influences proper completion of meiosis we examined kinetics of meiotic nuclear divisions and spore viability in the strains harboring the variants of GFP-Pch2 with different subcellular and chromosomal distributions. Consistent with the fact that, in otherwise unperturbed meiosis, and at least in the BR strain background, the absence of *PCH2* (*pch2Δ*) does not significantly affect meiotic progression, the kinetics of meiotic divisions of *GFP-NES-PCH2* and *GFP-NLS-PCH2* strains (in a *ZIP1* background) were almost indistinguishable to that of the wild-type *GFP-PCH2* strain (Fig 5A, top graph), despite having altered Pch2 localization (see above). Although Pch2 is involved in crossover control [58], spore viability is high in the *pch2Δ* single mutant (Fig 5B, top graph). Accordingly, spore viability was not largely affected when the localization of Pch2 is altered (Fig 5B, top graph). Nevertheless, the influence of Pch2 in crossover homeostasis can be unveiled in situations, such as in *spo11* hypomorph mutants, where global DSB levels are reduced [58,59]. Thus, we combined *GFP-PCH2*, *GFP-NES-PCH2* and *GFP-NLS-PCH2* with the *spo11-3HA* allele that confers about 80% of total DSB levels [60]. As reported, spore viability decreased in the *pch2Δ spo11-3HA* double mutant, but it was normal in all hemizygous *GFP-PCH2*, *GFP-NES-PCH2* and *GFP-NLS-PCH2* strains harboring *spo11-3HA*. However, increased nuclear accumulation of Pch2 in the *GFP-NLS-PCH2* homozygous strain led to reduced spore viability in combination with *spo11-3HA* (Fig 5B, top graph). Like in *spo11-3HA pch2Δ*, the pattern of spore death in the *spo11-3HA GFP-NLS-PCH2* homozygous strain showed a trend to the excess of tetrads with four-, two- and zero-viable spores, indicative of meiosis I nondisjunction events (Fig 5B, bottom graph; [58]). Thus, curiously, both the complete lack of Pch2 or its forced strong accumulation inside the nucleus lead to the same pathological meiotic outcome.

We next studied the functionality of the different Pch2 versions in the context of checkpoint activation by *zip1Δ*. The *zip1Δ GFP-PCH2* diploid displayed a tight meiotic arrest (Fig 5A,

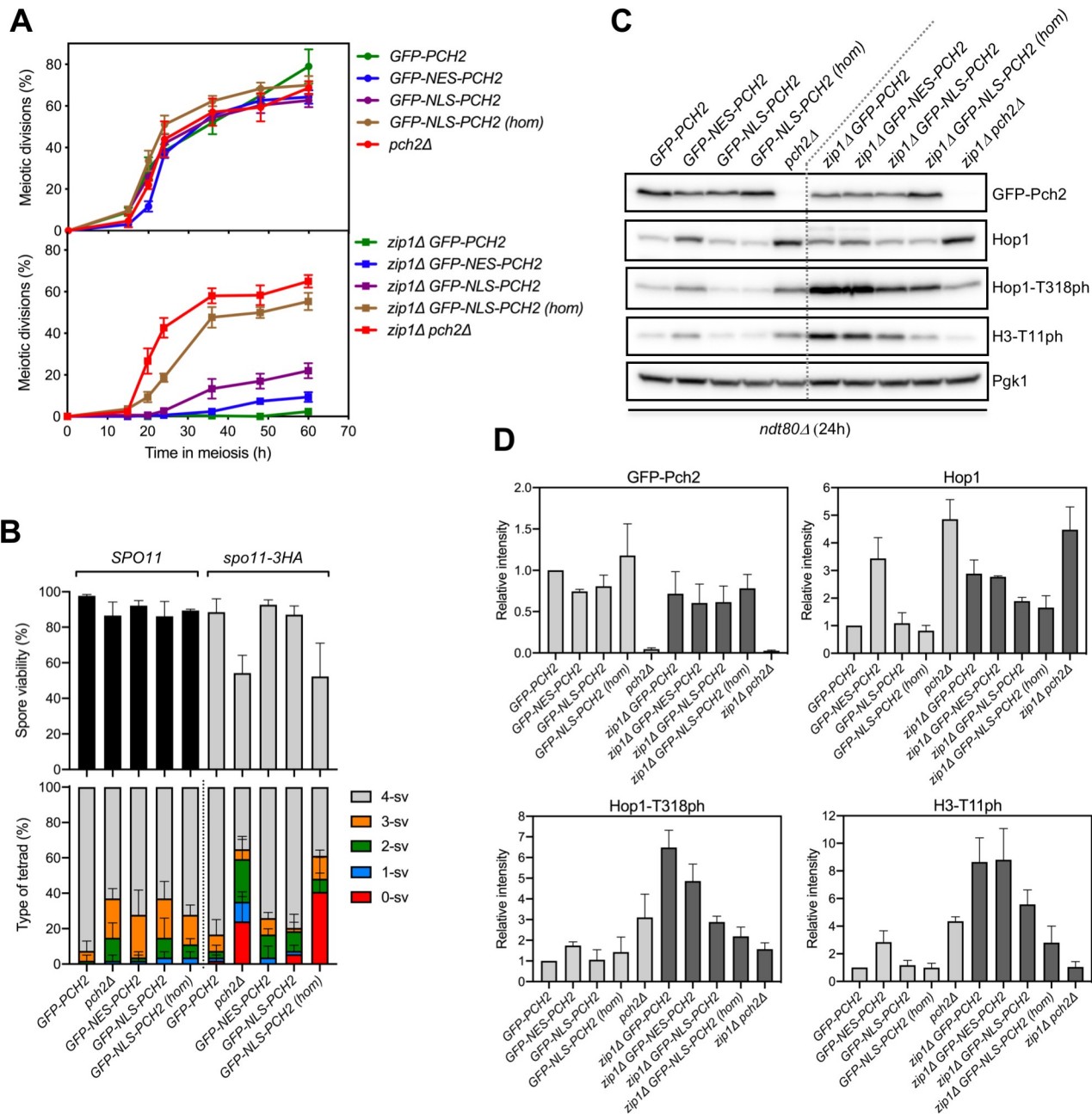

**Fig 5. The largely cytoplasmic GFP-NES-Pch2 version supports checkpoint function.** (**A**) Time course analysis of meiotic nuclear divisions; the percentage of cells containing two or more nuclei is represented. Error bars: SD; n = 3. At least 300 cells were scored for each strain at every time point. (**B**) Spore viability assessed by tetrad dissection is shown in the top graph. The percentage of tetrads containing 4-, 3-, 2-, 1-, and 0-viable spores is presented in the bottom graph. At least 54 tetrads were dissected for each strain. Error bars, SD; n = 3. (**C**) Representative western blot analysis of GFP-Pch2 and Hop1 production, and Hop1-T318 and H3-T11 phosphorylation (ph), in *ndt80Δ*-arrested strains of the indicated genotypes. Cell extracts were prepared at 24 hours in meiosis. (**D**) Quantification of GFP-Pch2, Hop1, Hop1-T318ph and H3-T11ph relative levels analyzed as in (C). The intensity values for each protein in each strain were normalized to Pgk1 and relativized to those of the wild-type *GFP-PCH2* strain. Error bars, SD; n = 3. Light grey and dark grey bars correspond to *ZIP1* and *zip1Δ* strains, respectively. Strains in (**A**) are: DP1624 (*GFP-PCH2*), DP1685 (*GFP-NES-PCH2*), DP1699 (*GFP-NLS-PCH2*), DP1695 (*GFP-NLS-PCH2 [hom]*), DP1023 (*pch2Δ*), DP1625 (*zip1Δ GFP-PCH2*), DP1686 (*zip1Δ GFP-NES-PCH2*), DP1701 (*zip1Δ GFP-NLS-PCH2*), DP1696 (*zip1Δ GFP-NLS-PCH2 [hom]*) and DP1029 (*zip1Δ pch2Δ*). Strains in (B) are: DP1624 (*GFP-PCH2*), DP1023 (*pch2Δ*), DP1685 (*GFP-NES-PCH2*), DP1699 (*GFP-NLS-PCH2*), DP1695 (*GFP-NLS-PCH2 [hom]*), DP1789 (*spo11-3HA GFP-PCH2*), DP1787 (*spo11-3HA pch2Δ*), DP1791 (*spo11-3HA GFP-NES-PCH2*), DP1793 (*spo11-3HA GFP-NLS-PCH2*) and DP1792 (*spo11-3HA GFP-NLS-PCH2 [hom]*). Strains in (C, D) are: DP1654 (*ndt80Δ GFP-PCH2*), DP1725 (*ndt80Δ GFP-NES-PCH2*), DP1729 (*ndt80Δ GFP-NLS-PCH2*), DP1768 (*ndt80Δ GFP-NLS-PCH2 [hom]*), DP1058 (*ndt80Δ pch2Δ*), DP1655 (*ndt80Δ zip1Δ GFP-PCH2*), DP1726 (*ndt80Δ zip1Δ GFP-NES-PCH2*), DP1730 (*ndt80Δ zip1Δ GFP-NLS-PCH2*), DP1769 (*ndt80Δ zip1Δ GFP-NLS-PCH2 [hom]*) and DP881 (*ndt80Δ zip1Δ pch2Δ*).

bottom graph) further corroborating the notion that GFP-Pch2 is fully functional. Remarkably, the meiotic block triggered by the absence of *ZIP1* was almost fully maintained in *zip1Δ GFP-NES-PCH2* (Fig 5A, bottom graph) that largely lacks nuclear Pch2 and exhibits a predominantly cytoplasmic Pch2 localization. In turn, the meiotic arrest was released to some extent in the hemizygous *zip1Δ GFP-NLS-PCH2* mutant indicative of a somewhat weaker checkpoint response. Moreover, higher levels of nuclear Pch2 achieved in the homozygous *zip1Δ GFP-NLS-PCH2* strain led to a strong checkpoint defect, as manifested by the substantial alleviation of the meiotic arrest, almost equivalent to that of *zip1Δ pch2Δ* (Fig 5A, bottom graph).

Due to the different kinetics of meiotic progression conferred by the different GFP-Pch2 versions, particularly in a *zip1Δ* background (Fig 5A, bottom graph), we used prophase-arrested *ndt80Δ* strains for an accurate quantification of protein levels of Pch2, Hop1 and checkpoint markers (Fig 5C and 5D). We have previously reported that the critical function of Pch2 in the checkpoint triggered by defective synapsis is to sustain proper levels of Mec1-dependent Hop1-T318 phosphorylation required for the ensuing Mek1 activation [21]. Thus, checkpoint activity supported by Pch2 was monitored by western blot analysis of phospho-Hop1-T318. Also, histone H3-T11 phosphorylation was determined as a proxy for Mek1 activation [61]. Global levels of GFP-NES-Pch2 and GFP-NLS-Pch2 were only slightly reduced compared to GFP-Pch2 (Fig 5C and 5D), thus validating our localization studies (Fig 4). Like in the *pch2Δ* single mutant, Hop1 was more abundant in the *GFP-NES-PCH2* strain, compared to *GFP-PCH2* suggesting that the nuclear exclusion of Pch2 may lead to increased Hop1 protein stability on synapsed chromosomes. However, unlike *zip1Δ pch2Δ*, Hop1 global levels were similar in *zip1Δ GFP-PCH2* and *zip1Δ GFP-NES-PCH2* strains, suggesting than in the context of unsynapsed chromosomes, nuclear exclusion of Pch2 has no effect on Hop1 stability. On the other hand, forced nuclear localization of Pch2 (GFP-NLS-Pch2) resulted in a modest reduction of total Hop1 protein, primarily, in *zip1Δ* cells. Quantitative analysis of the levels of checkpoint markers (Hop1-T318 and H3-T11 phosphorylation) demonstrated that checkpoint activity was largely maintained in *zip1Δ GFP-NES-PCH2*, was reduced in *zip1Δ GFP-NLS-PCH2* hemizygous strains and was further compromised in *zip1Δ GFP-NLS-PCH2* homozygous strains (Fig 5C and 5D, dark grey bars).

Therefore, we conclude that the amount and distribution of Pch2 inside the nucleus must be carefully balanced to avoid deleterious effects on meiosis and, mainly, on the checkpoint response to defective synapsis (*zip1Δ*). In agreement with the analysis of the *orc1-3mAID* mutant (see above), these results also confirm that the cytoplasmic pool of Pch2 is proficient to sustain its checkpoint activation function.

## Dot1 is irrelevant for the checkpoint when Pch2 is outside the nucleus

The histone H3K79 methyltransferase Dot1 is required for the checkpoint induced by the lack of Zip1; the absence of *DOT1* suppresses the meiotic block of *zip1Δ*. Furthermore, deletion of *DOT1* (or mutation of H3K79) results in delocalization of Pch2 from the nucleolus and its general distribution throughout chromatin [20,31]. These observations initially led to the hypothesis that the nucleolar localization of Pch2 may be important for its checkpoint function. However, we now have revealed a novel functionally-relevant cytoplasmic localization of Pch2 and we have demonstrated that the presence of Pch2 in the nucleolus is actually dispensable for the checkpoint ([56]; this work). Thus, to further delineate the functional impact of Dot1 action on Pch2 localization and checkpoint activity we analyzed the effect of deleting *DOT1* in those conditions where the checkpoint is still active but Pch2 is localized outside the nucleolus as a consequence of either Orc1 depletion (*zip1Δ orc1-3mAID GFP-PCH2*) or Pch2 fusion to a NES (*zip1Δ GFP-NES-PCH2*). Since the influence of Dot1 on Pch2 localization has been only

studied using chromosome spreads, we examined wild-type GFP-Pch2 subcellular distribution in the absence of *DOT1*. We found that, consistent with previous reports, GFP-Pch2 relocalized from the nucleolus to pan-nuclear foci upon *DOT1* deletion in *zip1Δ* cells (Fig 6A). However, mutation of *DOT1* had no effect on the cytoplasmic distribution of Pch2 in *zip1Δ orc1-3mAID GFP-PCH2* or *zip1Δ GFP-NES-PCH2* cells; that is, Pch2 remained in the cytoplasm in those conditions (Fig 6A and 6B). To assess the status of checkpoint activity in these strains we analyzed sporulation efficiency, kinetics of meiotic nuclear divisions, and phosphorylation of Hop1-T318 and H3-T11. As expected, deletion of *DOT1* suppressed the sporulation arrest of *zip1Δ GFP-PCH2*. However, the *zip1Δ orc1-3mAID GFP-PCH2 dot1Δ* and *zip1Δ GFP-NES-PCH2 dot1Δ* mutants did not sporulate, suggesting that the checkpoint remains active in these strains (Fig 6C). Moreover, like in *zip1Δ GFP-PCH2* and *zip1Δ orc1-3mAID GFP-PCH2*, meiotic progression in *zip1Δ orc1-3mAID GFP-PCH2 dot1Δ* was also completely blocked (Fig 6D), and high levels of Hop1-T318 and H3-T11 phosphorylation were maintained (Fig 6E), indicative of a robust checkpoint response. Nuclear divisions were also considerably delayed in *zip1Δ GFP-NES-PCH2 dot1Δ*, although a fraction of the cells resumed meiotic divisions at late time points (Fig 6D), and Hop1-T318 and H3-T11 phosphorylation also eventually declined (Fig 6E). This weaker checkpoint arrest in *zip1Δ GFP-NES-PCH2 dot1Δ* likely stems from the fact that a small amount of GFP-NES-Pch2 remains in the nucleolus in a fraction of cells expressing *GFP-NES-PCH2* (Figs 3A and S3). Mislocalization and widespread distribution of this residual nucleolar GFP-NES-Pch2 population upon *DOT1* deletion would lead to the eventual partial loss of checkpoint strength.

We conclude that when Pch2 is depleted from the nucleolus and accumulates in the cytoplasm, Dot1 is no longer required to support meiotic checkpoint activity. Our results also indicate that the critical checkpoint function of Dot1-dependent H3K79 methylation is to maintain the nucleolar confinement of Pch2 to impede its pathological action on unsynapsed chromosomes. Thus, in terms of Pch2 localization, two requirements must be fulfilled in a *zip1Δ* mutant to elicit a proper checkpoint response: Pch2 must be present in the cytoplasm and the access of nuclear Pch2 to the chromosomes must be prevented. The latter is achieved by Dot1-dependent Pch2 restraint in the rDNA region.

## Nup2 is not required for activation of the meiotic recombination checkpoint

Another factor influencing Pch2 localization is the Nup2 nucleoporin. It has been recently shown that, in wild-type (*ZIP1*) nuclei, Nup2 promotes the chromosomal localization of Pch2; the *nup2Δ* mutant exhibits an increased accumulation of Pch2 in the rDNA region at the expense of the chromosomal Pch2 fraction, resulting in an altered regional distribution of Hop1 [37]. However, the impact of Nup2 on the subcellular localization of Pch2 and on the *zip1Δ*-induced checkpoint is not known. We first analyzed the localization of GFP-Pch2 in live meiotic cells lacking Nup2. Consistent with the previous report using chromosome spreading techniques [37], the GFP-Pch2 chromosomal foci were largely lost in the *nup2Δ* single mutant (S5A Fig). Also, an increase in the cytoplasmic fraction of GFP-Pch2 was observed in the absence of *NUP2*; this cytoplasmic accumulation was especially prominent in *zip1Δ nup2Δ* cells that also showed reduced nucleolar GFP-Pch2 signal compared to *zip1Δ* (S5A–S5C Fig). We then examined sporulation efficiency to assess checkpoint functionality. Albeit with reduced efficiency, the *nup2Δ* single mutant sporulated; however, in contrast to *zip1Δ pch2Δ*, sporulation was completely blocked in the *zip1Δ nup2Δ* double mutant (S5D Fig). Thus, the checkpoint triggered by the absence of *ZIP1* is fully active in the *nup2Δ* mutant, which shows a conspicuous cytoplasmic Pch2 localization.

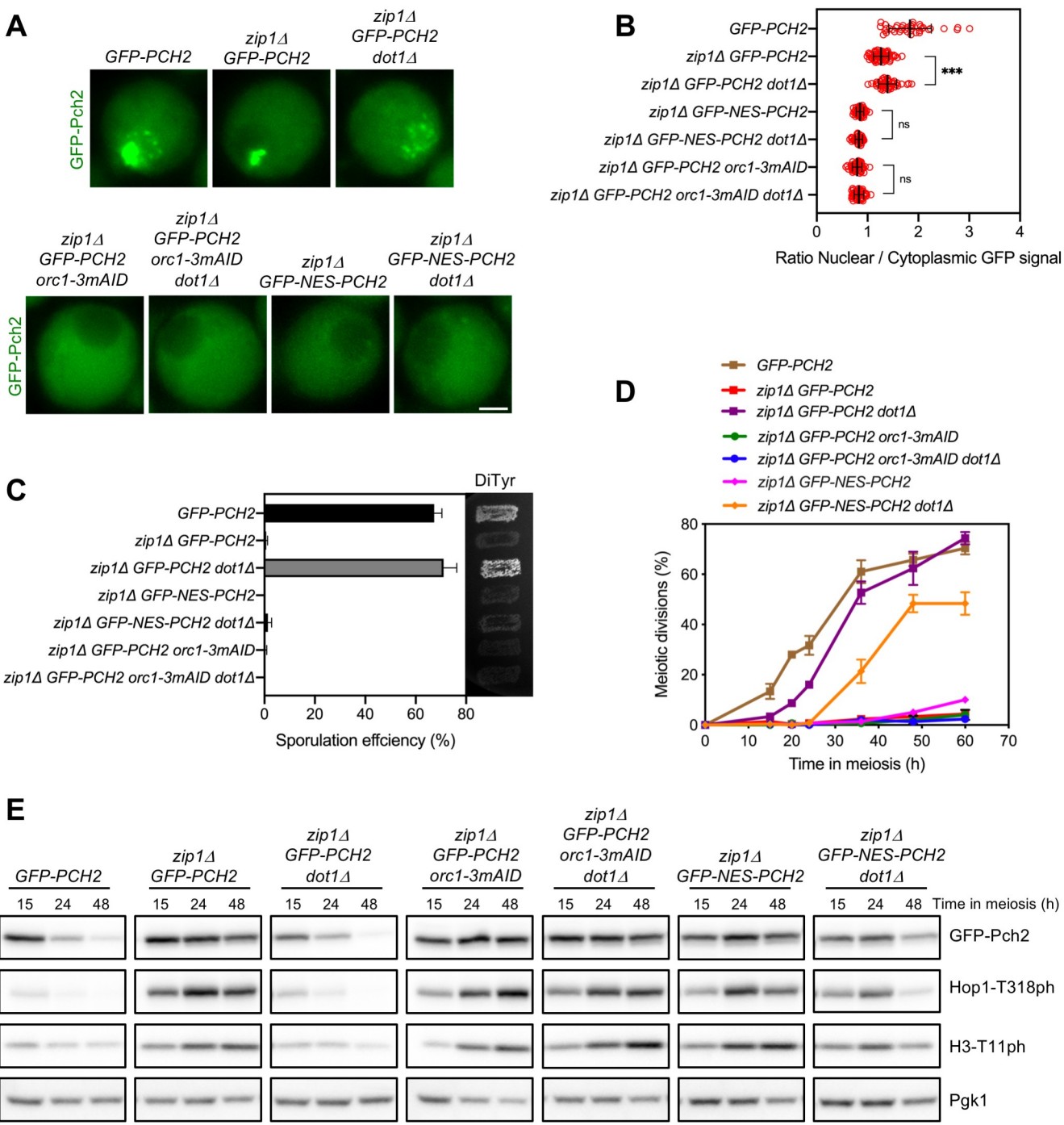

**Fig 6. Dot1 maintains Pch2 nucleolar confinement preventing its deleterious chromosomal binding. (A)** Fluorescence microscopy analysis of GFP-Pch2 distribution in whole meiotic cells of the indicated genotypes 15 hours after meiotic induction. Representative cells are shown. Scale bar, 2 μm. **(B)** Quantification of the ratio of nuclear (including nucleolar) to cytoplasmic mean GFP fluorescent signal in cells analyzed as in (A). Error bars: SD. **(C)** Sporulation efficiency, assessed by microscopic counting of asci, and dityrosine fluorescence (DiTyr), as a visual indicator of sporulation, were examined after 3 days on sporulation plates. Error bars, SD; n = 3. At least 300 cells were counted for each strain. **(D)** Time course analysis of meiotic nuclear divisions; the percentage of cells containing two or more nuclei is represented. Error bars: SD; n = 3. At least 300 cells were scored for each strain at every time point. **(E)** Western blot analysis of GFP-Pch2 production, and Hop1-T318 and H3-T11 phosphorylation (ph), at different meiotic time points. Pgk1 was used as a loading control. In all experiments, auxin (500 μM) was added 12 hours after meiotic induction to induce Orc1-3mAID depletion in cells harboring the degron allele [56]. Strains in (A-E) are: DP1624 (*GFP-PCH2*), DP1625 (*zip1Δ GFP- PCH2*), DP1734 (*zip1Δ GFP-PCH2 dot1Δ*), DP1644 (*zip1Δ GFP-PCH2 orc1-3mAID*), DP1746 (*zip1Δ GFP-PCH2 orc1-3mAID dot1Δ*), DP1686 (*zip1Δ GFP-NES-PCH2*) and DP1747 (*zip1Δ GFP-NES-PCH2 dot1Δ*).

## GFP-Pch2 tethering to the plasma membrane leads to constitutive checkpoint-dependent meiotic arrest

We have shown here that, in *zip1Δ orc1-3mAID GFP-PCH2* and *zip1Δ GFP-NES-PCH2* cells, the exclusive or preponderant, respectively, presence of Pch2 in the cytoplasm is sufficient to sustain meiotic checkpoint function. To further reinforce this notion and to explore whether Pch2 requires to be freely diffusible in the cytoplasm to exert its action, we took advantage of the GFP-binding protein (GBP) to tether GFP-NES-Pch2 (or GFP-Pch2) to a fixed cellular location outside the nucleus. In particular, we used a Pil1-GBP-mCherry fusion protein to force the localization of GFP-NES-Pch2 (or GFP-Pch2) to the eisosomes, which are immobile protein assemblies located at specialized domains of the plasma membrane. Pil1 is a major subunit of the eisosomes positioned at the cytoplasmic surface of the plasma membrane [62,63]. We generated *ZIP1* and *zip1Δ* strains harboring both *PIL1-GBP-mCherry* and *GFP-NES-PCH2*. As reported for eisosome localization, Pil1-GBP-mCherry formed quite uniform patches decorating the cellular periphery (Fig 7A). Remarkably, the GFP-NES-Pch2 protein was efficiently driven to the plasma membrane compartment containing Pil1-GBP-mCherry, displaying a robust colocalization (Fig 7A and 7B). Not only the largely cytoplasmic GFP-NES-Pch2 protein was recruited to Pil1-GBP-mCherry patches; also, the wild-type GFP-Pch2 version was completely moved from its nuclear/nucleolar localization to the plasma membrane (S6A Fig). Highlighting the efficient sequestration of Pch2 at the plasma membrane, the Hop1 protein, which is normally excluded from the rDNA by Pch2 (Figs 4Af and 7Ca), was conspicuously present in this region (identified by the nucleolar Nsr1 protein) in spread nuclei of *zip1Δ GFP-NES-PCH2 PIL1-GBP-mCherry* (Fig 7Cc) and *GFP-PCH2 PIL1-GBP-mCherry* (S6B Fig). Like in *zip1Δ GFP-PCH2* and *zip1Δ GFP-PCH2-NES* (Fig 7Ca-7Cb), Hop1 also displayed a continuous and even stronger localization along unsynapsed axes in *zip1Δ GFP-NES-PCH2 PIL1-GBP-mCherry* (Fig 7Cc and 7D), contrasting with *zip1Δ pch2Δ* in which Hop1 linear localization is impaired ([21]; Fig 7Cd and 7D).

We next examined sporulation efficiency to assess checkpoint functionality. Like in *zip1Δ GFP-PCH2* and *zip1Δ GFP-NES-PCH2*, sporulation was blocked in the *zip1Δ GFP-NES-PCH2 PIL1-GBP-mCherry* strain (Fig 7E), indicating that the checkpoint is still active when Pch2 is anchored to the plasma membrane. Accordingly, high levels of Hop1-T318 and H3-T11 phosphorylation were maintained in *zip1Δ GFP-NES-PCH2 PIL1-GBP-mCherry* (Fig 7F). Unexpectedly, sporulation was also arrested, and high levels of active checkpoint markers were also achieved in otherwise wild-type cells (i.e., *ZIP1*) harboring *PIL1-GBP-mCherry* together with either *GFP-NES-PCH2* (Fig 7E and 7F) or *GFP-PCH2* (S6C Fig). This sporulation block was relieved by deletion of *MEK1* or *SPO11* (Figs 7E, S6C and S6D) demonstrating that it resulted from activation of the meiotic recombination checkpoint. Thus, these results indicate not only that the checkpoint function of Pch2 is imposed from outside the nucleus, but also that immobilization of Pch2 in a fixed extranuclear compartment, namely the plasma membrane, leads to constitutive checkpoint activation. Furthermore, these observations also imply that Pch2 does not need to be freely diffusible in the cytoplasm to gain access to its substrate.

## Discussion

In this work we use different strategies to manipulate Pch2 localization within meiotic prophase I cells to establish the biological relevance, particularly for checkpoint function, of the presence of Pch2 in the different compartments where it can be located: unsynapsed rDNA region, synapsed chromosomes and cytoplasm. Since most, if not all, known meiotic events influenced by Pch2 activity occur in the nucleus, and the main Pch2 substrate, Hop1, is a component of chromosome axes, the majority of previous localization studies of Pch2 (and the

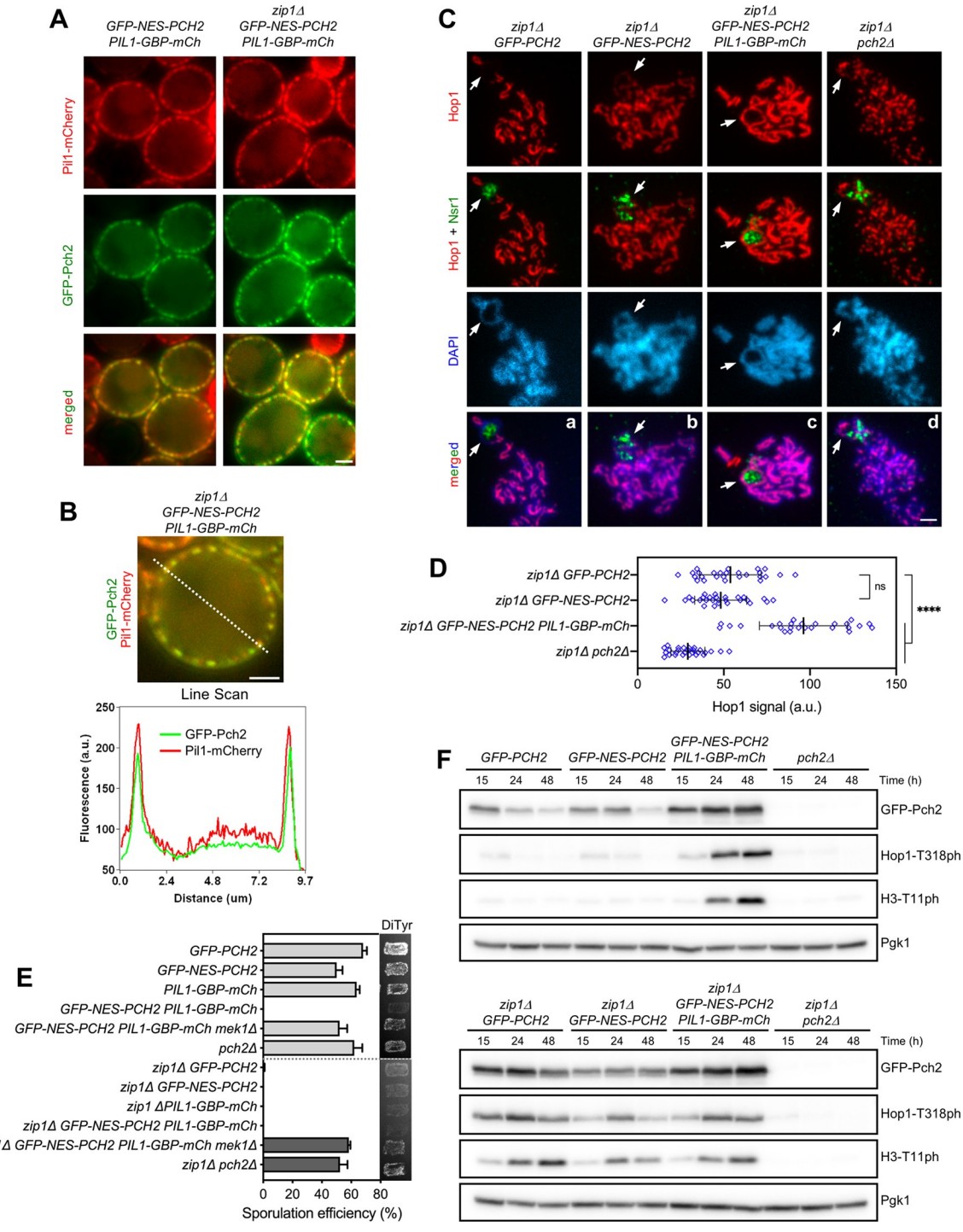

**Fig 7. Immobilization of Pch2 at the cell periphery leads to constitutive checkpoint induction. (A)** Fluorescence microscopy analysis of GFP-Pch2 and Pil1-GBP-mCherry distribution in whole meiotic cells of the indicated genotypes 16 hours after meiotic induction. Scale bar, 2 μm. **(B)** A representative cell and the corresponding Line Scan plot are shown to highlight GFP-Pch2 and Pil1-GBP-mCherry colocalization. The graph represents the GFP and mCherry fluorescent signals (green and red, respectively) along the depicted dotted line from left to right (a.u., arbitrary units). Scale bar, 2 μm. **(C)** Immunofluorescence of meiotic chromosomes stained with anti-Hop1 (red) and anti-Nsr1 (nucleolar marker; green) antibodies, and DAPI (blue). Arrows point to the rDNA region. Spreads were prepared at 16 h. Scale bar, 2 μm. **(D)** Quantification of the Hop1 signal on spreads. Error bars: SD; a.u., arbitrary units. **(E)** Sporulation efficiency and dityrosine fluorescence (DiTyr) were examined after 3 days on sporulation plates. Error bars, SD; n = 3. At least 300 cells were counted for each strain. **(F)** Western blot analysis of GFP-Pch2 production and checkpoint activation markers (Hop1-T318 and H3-T11 phosphorylation), at different meiotic time points. Pgk1 was used as a loading control. Strains in (A-E) are: DP1624

(*GFP-PCH2*), DP1685 (*GFP-NES-PCH2*), DP1802 (*PIL1-GBP-mCherry*), DP1795 (*GFP-NES-PCH2 PIL1-GBP-mCherry*), DP1811 (*GFP-NES-PCH2 PIL1-GBP-mCherry mek1Δ*), DP1023 (*pch2Δ*), DP1625 (*zip1Δ GFP-PCH2*), DP1686 (*zip1Δ GFP-NES-PCH2*), DP1803 (*zip1Δ PIL1-GBP-mCherry*), DP1796 (*zip1Δ GFP-NES-PCH2 PIL1-GBP-mCherry*), DP1812 (*zip1Δ GFP-NES-PCH2 PIL1-GBP-mCherry mek1Δ*) and DP1029 (*zip1Δ pch2Δ*).

orthologs in other organisms) have been exclusively focused on its chromosomal distribution. However, we have recently revealed that Pch2 also shows a diffuse cytoplasmic localization [56]. Here, we demonstrate that the presence of Pch2 in the cytoplasm is essential for the checkpoint response to the absence of *ZIP1* and define the functional contribution of Pch2 regulators such as Orc1, Dot1 and Nup2 for a balanced distribution of Pch2 in different subcellular compartments. The most relevant observations relating Pch2 localization with Hop1 chromosomal pattern and checkpoint function in the different conditions analyzed are compiled in Table 1. A model for Pch2 action in the meiotic recombination checkpoint is presented in Figs 8 and S7.

## Activation of the meiotic recombination checkpoint relies on cytoplasmic Pch2

We have taken advantage of a functional version of Pch2 tagged with GFP to dissect the localization of Pch2 in whole meiotic cells. We have first analyzed GFP-Pch2 distribution upon Orc1 depletion. Orc1 recruits Pch2 to the rDNA region [51,56], but the absence of Orc1 does not alter the checkpoint response induced by *zip1Δ* indicating that the accumulation of Pch2 in the nucleolar region is only required to prevent recombination in the rDNA array [31,51], but it is dispensable for checkpoint activation [56]. We show that GFP-Pch2 is only detected in the cytoplasm of *zip1Δ* meiotic cells lacking Orc1 indicating that Pch2 is capable of orchestrating the checkpoint response from this extra-nuclear location. Recent work using chromatin immunoprecipitation has proposed that, besides the rDNA, Orc1 also promotes Pch2 binding to a subset of RNA polymerase II-dependent actively transcribed genes [64]. The biological

**Table 1. Summary of Pch2 subcellular localization, Hop1 chromosomal distribution and checkpoint activity in different situations.**

| Relevant Genotype[1] | Pch2 subcellular localization | | | | Hop1 localization | | Checkpoint Activity |
|---|---|---|---|---|---|---|---|
| | Chromosomes | Nucleolus (rDNA) | Cytoplasm | Other | Chromosome axes | Nucleolus (rDNA) | |
| wild type | + (faint foci) | + | + | - | foci | - | na |
| *zip1Δ* | - | + | + | - | linear | - | + |
| *zip1Δ pch2Δ* | - | - | - | - | discontinuous | + | - |
| *zip1Δ orc1-3mAID* | - | - | + | - | linear[2] | +[2] | + |
| *zip1Δ NES-PCH2* | - | -/+ | + | - | linear | + | + |
| *zip1Δ NLS-PCH2 (hem)* | - | + | -/+ | nucleoplasm | discontinuous | - | -/+ |
| *zip1Δ NLS-PCH2 (hom)* | - | + | -/+ | nucleoplasm | discontinuous | - | - |
| *zip1Δ dot1Δ* | + (widespread) | + | + | - | discontinuous[3] | +/-[3] | - |
| *zip1Δ orc1-3mAID dot1Δ* | - | - | + | - | nd | nd | + |
| *zip1Δ nup2Δ* | - | +/- | + | - | nd | nd | + |
| *zip1Δ PIL1-GBP* | - | - | - | eisosomes | strong linear | + | + |

(1) Strains carry GFP-tagged *PCH2*.

(2) Data from [56].

(3) Data from [20].

na: not applicable; nd: not determined.

*hem*: hemizygous; *hom*: homozygous

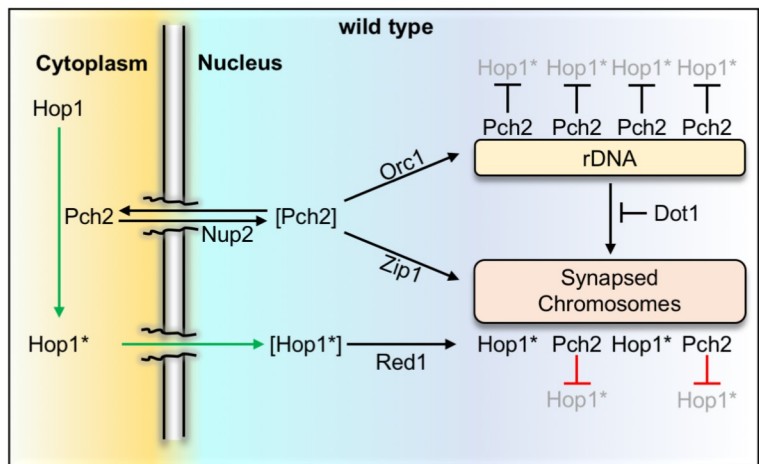

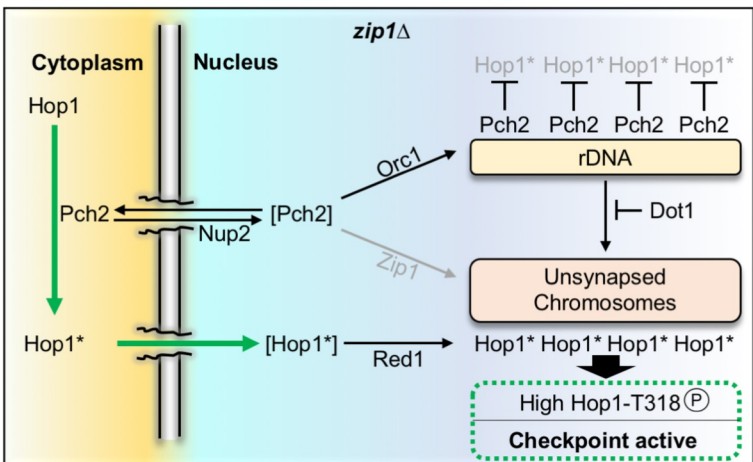

**Fig 8. Model of Pch2 function in the meiotic recombination checkpoint induced by the lack of Zip1.** The population of Pch2 present in the cytoplasm acts on Hop1 to provoke a conformational change that poises Hop1 (indicated by an asterisk) for its transport to the nucleus and the incorporation on chromosomes via Red1 interaction. Initially, this predisposed state, Hop1*, likely arises by the conversion from the closed to unbuckled conformation, although additional mechanisms may be also involved. The precise distribution of Pch2 among cytoplasm, rDNA and synapsed chromosomes is carefully balanced by the factors depicted. In the wild type, Zip1-mediated recruitment of Pch2 to synapsed chromosomes leads to Hop1 eviction likely by releasing the interaction with Red1. In the *zip1Δ* mutant, Pch2 does not localize to chromosomes and the cytoplasmic pool is increased further stimulating Hop1 incorporation on the unsynapsed axes and the subsequent Mec1-dependent phosphorylation at T318 to launch the downstream checkpoint activation response. Proteins between brackets represent transient states. See text and S7 Fig for additional details and model of action in different circumstances.

relevance of this additional pool of euchromatin-associated Pch2 remains to be established, but several lines of evidence indicate that it is not involved in the *zip1Δ*-induced meiotic recombination checkpoint. First, recruitment of Pch2 to these transcribed genes is largely diminished in the absence of *ZIP1* [64], implying that another Pch2 population must perform the *zip1Δ*-induced checkpoint activation task. Second, the checkpoint is intact in the absence of Orc1, therefore, it does not rely on Orc1-mediated recruitment of Pch2 to anywhere in the genome, either heterochromatin (rDNA) or euchromatin (RNA polymerase II-transcribed genes). Third, although the crucial checkpoint role of Pch2 is to sustain Hop1-T318 phosphorylation [21], the binding of Pch2 to the body of actively transcribed genes has no impact

whatsoever on Hop1 localization [64]. Importantly, the checkpoint arrest of *zip1Δ orc1-3mAID* is alleviated by *PCH2* deletion [56] demonstrating that it does not stem from the activation of another independent meiotic surveillance mechanism and supporting the notion that Pch2 is still required for the checkpoint in this scenario acting from the cytoplasmic localization.

To further confirm the localization requirements of Pch2 for checkpoint activity we forced the accumulation of GFP-Pch2 inside or outside the nucleus by fusion to ectopic canonical NLS or NES sequences, respectively. Checkpoint function measured by various parameters is largely maintained in *zip1Δ GFP-NES-PCH2* cells that display a predominant cytoplasmic localization of Pch2 (S7C Fig). Although the majority of the GFP-NES-Pch2 protein is impelled towards the cytoplasm, some remnants can be still detected inside the nucleus, especially in the nucleolus, likely accounting for the slightly weaker checkpoint arrest observed in *zip1Δ GFP-NES-PCH2*, compared to the absolute block of *zip1Δ orc1-3mAID* in which no hint of GFP-Pch2 is detected inside the nucleus. Conversely, NLS-mediated buildup of GFP-Pch2 inside the nucleus leads to checkpoint inactivation. Recent work has proposed that Pch2 possesses both a non-chromosomal checkpoint activating function and a chromosomal-dependent checkpoint-silencing function [14]. Our results indicate that the non-chromosomal activating role of Pch2 is actually established in the cytoplasm. Moreover, we show that despite the large amount of Pch2 protein accumulated inside the nucleus in homozygous *zip1Δ GFP-NLS-PCH2* strains, it is only gathered in the nucleolus and also in the nucleoplasm, but it is not associated to (non-rDNA) unsynapsed chromosomes (Figs 3 and 4). This observation raises the possibility that the detrimental effect of nuclear Pch2 on checkpoint activity could be exerted even from the nucleoplasm (S7D Fig). Interestingly, the pathological effect of excessive nuclear Pch2 is not exclusive of the checkpoint response to unsynapsed chromosomes; in *ZIP1* strains harboring a hypomorphic *spo11-3HA* allele conferring reduced DSB levels, nuclear accumulation of GFP-NLS-Pch2 is also deleterious (Fig 5B), although in this case the effect may well be resulting from unrestrained Zip1-mediated recruitment of Pch2 to chromosomes.

## Dot1 is required for Pch2 nucleolar confinement

Dot1-dependent H3K79me is required for the *zip1Δ* meiotic block triggered by the checkpoint [20,54]. Surprisingly, we demonstrate here that when Pch2 is exclusively located outside the nucleus (i.e., in a *zip1Δ orc1-3mAID* mutant), Dot1 is no longer needed for the checkpoint. Since Pch2 loses the nucleolar confinement in *zip1Δ dot1Δ* cells and it is found throughout chromatin (Fig 6A; [20,54]), the simplest interpretation is that Dot1 is mainly required to maintain Pch2 sequestered in the rDNA chromatin preventing its nuclear dispersion and the subsequent negative action on checkpoint activity of widespread nuclear Pch2 (S7E and S7F Fig). We note that in the *zip1Δ dot1Δ* double mutant, Pch2 is capable of binding to unsynapsed chromosomes indicating that, in the absence of H3K79me, Pch2 can be recruited to chromosomes independently of Zip1. In fact, Pch2 is naturally recruited to the rDNA region, which is devoid of Zip1. We propose that global H3K79 methylation, likely full H3K79me3 [20], limits general Pch2 recruitment and that low levels of H3K79me in the rDNA allow Pch2 binding thus ensuring Pch2 confinement to the nucleolar region. Dot1 activity is stimulated by H4K16 acetylation [65]. In prophase I meiotic nuclei, H4K16ac is widely distributed throughout chromatin, but it is excluded from the rDNA area [53], thus supporting the notion that Dot1-dependent H3K79me should be reduced in the rDNA lacking H4K16ac. Pch2 binding to sites of active transcription shows some degree of correlation with H3K79me1 and it has been suggested that this modification may contribute to Orc1-dependent recruitment of Pch2 [64]. Since the H3K79me1 isoform results from limited Dot1 catalytic activity [66]; this observation may still be in agreement with the idea that full Dot1 enzymatic activity producing high levels

of H3K79me3 prevents Pch2 binding. However, we note that the Orc1-driven transcription-associated Pch2 population is not involved in checkpoint regulation (see above), and that Pch2 is prominently recruited to unsynapsed chromosomes in the *dot1Δ* mutant lacking all forms of H3K79me, including H3K79me1.

### Nup2 may promote Pch2 nuclear import

Nup2 is a mobile nucleoporin located in the basket of nuclear pore complexes (NPCs) that controls Hop1 retention in the so-called chromosome end-adjacent regions (EARs), which sustain continued meiotic DSB formation even after the establishment of the SC. Nup2 modulates Hop1 retention in the EARs by regulating Pch2 chromosomal distribution. Using chromosome spreading, it has been shown that, in the *nup2Δ* mutant, Pch2 chromosomal foci are greatly diminished, and most Pch2 concentrates in the nucleolar region [37]. While our analysis of GFP-Pch2 localization in whole meiotic cells also reveals the absence of Pch2 chromosomal signal in *nup2Δ*, we do not detect a significant increase in the nucleolar signal (S5C Fig). This apparent discrepancy may result from the different detection technique or from certain peculiarities of the different strain background used. Indeed, the relative intensity of chromosomal and nucleolar Pch2 signal varies between wild-type BR and SK1 strains, with the chromosomal foci more easily detectable in SK1 [34,50] and the nucleolar accumulation more prominent in BR [21,56]. On the other hand, our results clearly show an increased cytoplasmic retention of GFP-Pch2 in the absence of *NUP2*; this cytoplasmic accumulation is particularly pronounced in the *zip1Δ nup2Δ* double mutant (S5A Fig). Consistent with the notion that the contribution of Pch2 to checkpoint activation in *zip1Δ* is established from the cytoplasm, sporulation is completely blocked in *zip1Δ nup2Δ* indicative of checkpoint proficiency. Nup2 contains a denominated "meiotic autonomous region" that, in addition to the nuclear periphery, also localizes to foci on meiotic chromosomes [67]. However, it is unlikely that Nup2 exerts a direct local control of Pch2 chromosomal distribution because, in *zip1Δ* cells, Pch2 is not recruited to chromosomes, but Pch2 subcellular distribution is still altered in the absence of *NUP2*. Since Nup2 is involved in the nuclear import of proteins in vegetative cells [68] and we find a cytoplasmic accumulation of Pch2 in *nup2Δ* mutants, we speculate that this nucleoporin may facilitate the entry and/or release of Pch2 into the nucleus via NPCs (Figs 8 and S7G). Nevertheless, Nup2 is also involved in spatial organization of the genome [69]; therefore, an alternative or additional mode of Pch2 regulation by Nup2 within the nucleus, likely not involved in checkpoint activation, cannot be excluded.

### Nucleocytoplasmic communication underlies meiotic checkpoint function

Our work highlights the relevance of nucleocytoplasmic traffic for the biology of meiosis; particularly, for the correct functionality of the meiotic recombination checkpoint. We propose that biochemical events occurring outside the nucleus have an impact on chromosomal transactions, namely Hop1 localization and phosphorylation. Since the only known substrate of Pch2 is the Hop1 protein, and the catalytic activity of Pch2 is required for the checkpoint [21], we postulate that Pch2 exerts an ATPase-dependent conformational change on Hop1 in the cytoplasm that poises it for its transport inside the nucleus, perhaps by exposing a cryptic NLS, and the subsequent Red1-dependent incorporation and Mec1-dependent phosphorylation on unsynapsed axes of the *zip1Δ* mutant (Figs 8 and S7). Indeed, the C-terminus of Hop1, where the closure motif is located, contains a region rich in arginine and lysine residues that resembles an NLS. Moreover, SUMO-conjugated sites with a potential regulatory role have been recently identified in this Hop1 C-terminal region [70]. It is likely that the conformational change involves the transition from the closed to the unbuckled conformation of Hop1

described *in vitro* [44], but the occurrence of this event remains to be demonstrated *in vivo*. Recent studies in *Arabidopsis thaliana* have also proposed a role for PCH2 in the nuclear transport of ASY1 (the Hop1 homolog) and posit that PCH2 would perform a second conformational change inside the nucleus, specifically in the nucleoplasm, required for the incorporation of ASY1 to the axial elements during early prophase I [71]. We have shown that Hop1 axial localization is discontinuous in the *zip1Δ pch2Δ* mutant ([21]; Figs 4Aj and 7Cd), indicating that Pch2 is also required for efficient Hop1 incorporation on budding yeast unsynapsed chromosomes. However, we demonstrate here that the cytoplasmic Pch2 population is solely in charge of promoting efficient Hop1 loading: in *zip1Δ orc1-3mAID* and in *zip1Δ GFP-NES-PCH2 PIL1-GBP* strains, Pch2 is exclusively localized outside the nucleus, and Hop1 displays a robust chromosomal linear pattern. Moreover, our results are in line with a recent report invoking a checkpoint silencing role for chromosomal Pch2 [14], because we show that the aberrant accumulation of Pch2 inside the nucleus (NLS-Pch2) or its unscheduled widespread chromatin incorporation in *zip1Δ dot1Δ* lead to checkpoint defects. Our striking observation that the tight tethering of Pch2 to the inner face of the plasma membrane leads to a constitutively active checkpoint response even in a synapsis-proficient context underscores the notion that, in this situation, only the extranuclear checkpoint activating function of Pch2 is manifested, whereas the nuclear silencing action is completely absent. It is likely that Pch2 always conducts the same biochemical transaction, namely Hop1 conversion from closed to unbuckled conformation, but depending on the subcellular location it causes opposite effects. Thus, the balance of Pch2 subcellular distribution and the dynamic communication among the different compartments must be strictly controlled for a proper meiotic recombination checkpoint response.

We note that in an otherwise unperturbed meiosis, the *pch2Δ* single mutant loads Hop1 on chromosomes and displays high levels of spore viability, implying that there must be an equilibrium between both states of Hop1 and that, to a limited extent, Hop1 may inherently undergo the conformational change even without the participation of Pch2. In the *pch2Δ* mutant, the absence of the nuclear removal action exerted on Hop1 permits the incorporation of sufficient Hop1 to support rather normal meiosis.

We envision that the emerging cytoplasmic function of Pch2 in the meiotic recombination checkpoint may be evolutionarily conserved. In *C. elegans*, PCH-2 localizes to pachytene chromosomes suggesting a direct local role for PCH-2 in regulating recombination, likely by unlocking HORMAD proteins in collaboration with the CMT-1 cofactor [45,49]. However, worm PCH-2 is also required for the meiotic checkpoint induced in the synapsis-defective *syp-1* mutant [72]. In this scenario, PCH-2 does not localize to chromosomes, opening the possibility that the diffuse extranuclear PCH-2 signal observed in *syp-1* [73,74], analogous to the cytoplasmic distribution of budding yeast Pch2 characterized here, may be relevant for the checkpoint in worms.

Together, our work provides new insights into fundamental determinants for Pch2 localization among the different compartments where the protein performs specialized functions. Remarkably, a plethora of distinct cellular mechanisms, including chromatin modifications and topology, nuclear transport, and replication factors, influences Pch2 regulation. Further understanding the interconnections in the regulatory network orchestrating the precise balance of Pch2 subcellular distribution and how it impinges on Hop1 status to ensure accurate completion of critical meiotic events will be an intriguing future venture.

## Materials and methods

### Yeast strains

The genotypes of yeast strains are listed in S1 Table. All strains are in the BR1919 background [75]. The *zip1Δ::LEU2*, *zip1Δ::LYS2*, *ndt80Δ::LEU2*, *ndt80Δ::kanMX3*, *pch2Δ::URA3*, *pch2Δ::*

*TRP1*, *dot1Δ*::*URA3* and *dot1Δ*::*kanMX6* gene deletions were previously described [20,21,54]. The *mek1Δ*::*natMX4* and *nup2Δ*::*hphMX4* deletions were made using a PCR-based approach [76]. Strains harboring the *spo11-3HA-6His*::*kanMX4* allele were obtained by transforming cells with a 2.2-kb *Eco*RI-*Sac*II restriction fragment from pSK54 [77]. N-terminal tagging of Pch2 with three copies of the HA epitope, *HOP1-mCherry* tagging, and the *orc1-3mAID* construct have been previously described [31,56,78].

The $P_{HOP1}$-*GFP-PCH2* construct [56], as well as $P_{HOP1}$-*GFP-NES-PCH2* and $P_{HOP1}$-*GFP-NLS-PCH2*, were introduced into the genomic locus of *PCH2* using an adaptation of the *delitto perfetto* technique [79]. Basically, PCR fragments flanked by the appropriate sequences were amplified from pSS393, pSS408 or pSS421 (see below), containing the *HOP1* promoter followed by the *GFP*, *GFP-NES* or *GFP-NLS* sequences, respectively, and a five Gly-Ala repeat linker before the second codon of *PCH2*. These fragments were transformed into a strain carrying the CORE cassette (*kanMX4-URA3*) inserted close to the 5' end of *PCH2*. G418-sensitive and 5-FOA-resistant clones containing the correct integrated construct, which results in the elimination of 91 nt of the *PCH2* promoter, were checked by PCR and verified by sequencing.

*PIL1-GBP-mCherry* strains were made following a normal PCR-based strategy for C-terminal tagging using a pFA6a-derived vector (pSS383) containing *GBP-mCherry*::*hphMX6*, kindly provided by A. Fernández-Álvarez (UPO, Sevilla).

All constructions and mutations were verified by PCR analysis and/or sequencing. The sequences of all primers used in strain construction are available upon request. All strains were made by direct transformation of haploid parents or by genetic crosses always in an isogenic background. Diploids were made by mating the corresponding haploid parents and isolation of zygotes by micromanipulation.

## Plasmids

The plasmids used are listed in S2 Table. The pSS393 centromeric plasmid expressing $P_{HOP1}$-*GFP-PCH2* was previously described [56]. The pSS408 and pSS421 plasmids driving the expression of $P_{HOP1}$-*GFP-NES-PCH2* and $P_{HOP1}$-*GFP-NLS-PCH2*, respectively, were derived from pSS393. An approximately 350 bp fragment corresponding to N-terminal region of *PCH2* was amplified from pSS393 with forward primers encoding the canonical NES (LALK-LAGLDI) [80] or NLS (PKKKRKV) [81] sequences preceded by a *Not*I site at the 5' end, and a reverse primer within the *PCH2* coding sequence downstream of the endogenous *Bam*HI site. These fragments were digested with *Not*I-*Bam*HI and cloned into the same sites of pSS393.

## Meiotic cultures and meiotic time courses

To induce meiosis and sporulation, BR strains were grown in 3.5 ml of synthetic complete medium (2% glucose, 0.7% yeast nitrogen base without amino acids, 0.05% adenine, and complete supplement mixture from Formedium at twice the particular concentration indicated by the manufacturer) for 20–24 h, then transferred to 2.5 ml of YPDA (1% yeast extract, 2% peptone, 2% glucose, and 0.02% adenine) and incubated to saturation for an additional 8 h. Cells were harvested, washed with 2% potassium acetate (KAc), resuspended into 2% KAc (10 ml), and incubated at 30°C with vigorous shaking to induce meiosis. Both YPDA and 2% KAc were supplemented with 20 mM adenine and 10 mM uracil. The culture volumes were scaled up when needed. To induce Orc1-3mAID degradation, auxin (500μM) was added to the cultures 12 h after meiotic induction.

To score meiotic nuclear divisions, samples from meiotic cultures were taken at different time points, fixed in 70% ethanol, washed in phosphate-buffered saline (PBS) and stained with 1 μg/μl 4′,6-diamidino-2- phenylindole (DAPI) for 15 min. At least 300 cells were counted at

each time point. Meiotic time courses were repeated several times; averages and error bars from at least three replicates are shown.

## Western blotting

Total cell extracts for Western blot analysis were prepared by trichloroacetic acid (TCA) precipitation from 5-ml aliquots of sporulation cultures, as previously described [27]. The antibodies used are listed in S3 Table. The ECL, ECL2 or SuperSignal West Femto reagents (ThermoFisher Scientific) were used for detection. The signal was captured on films and/or with a Fusion FX6 system (Vilber) and quantified with the Evolution-Capt software (Vilber).

## Cytology

Immunofluorescence of chromosome spreads was performed essentially as described [82]. The antibodies used are listed in S3 Table. Images of spreads were captured with a Nikon Eclipse 90i fluorescence microscope controlled with MetaMorph software (Molecular Devices) and equipped with a Hammamatsu Orca-AG charge-coupled device (CCD) camera and a PlanApo VC 100x 1.4 NA objective. Images of whole live cells expressing *GFP-PCH2*, *HOP1-mCherry* and *PIL1-GBP-mCherry* were captured with an Olympus IX71 fluorescence microscope equipped with a personal DeltaVision system, a CoolSnap HQ2 (Photometrics) camera, and 100x UPLSAPO 1.4 NA objective. Stacks of 7 planes at 0.8-μm intervals were collected. Maximum intensity projections of 3 planes containing Hop1-mCherry signal and single planes of GFP-Pch2 are shown in Figs 1C, 3A, 6A, S2A, S3 and S5A. In Figs 7A and S6A, a single plane of Pil1-mCherry and GFP-Pch2 is shown. The line-scan tool of the MetaMorph software was used to measure and plot the fluorescence intensity profile across the cell in Fig 7B. To determine the nuclear/cytoplasm GFP fluorescence ratio shown in Figs 1D, 3B, 6B and S5B, the ROI manager tool of Fiji software [83] was used to define the cytoplasm and nuclear (including the nucleolus) areas and the mean intensity values were measured. Similar results were obtained when the ratio of nuclear/cytoplasmic GFP signal was determined using total GFP fluorescence values within the area selected (S8 Fig). The Hop1-mCherry signal in whole cells was measured by using the same system but defining only the nuclear region. On the other hand, to determine the nucleolar GFP-Pch2 intensity, only the nucleolus was defined based on the conspicuous GFP-Pch2 structure restricted to one side of the nucleus. To determine Pch2 and Hop1 intensity on chromosome spreads, a region containing DAPI-stained chromatin was defined and the Raw Integrated Density values were measured. Background values were subtracted prior to ratio calculation. For background subtraction, the rolling ball algorithm from Fiji was used setting the radius to 50 pixels.

## Dityrosine fluorescence assay, sporulation efficiency, and spore viability

To examine dityrosine fluorescence as an indicator of the formation of mature asci, patches of cells grown on YPDA plates were replica-plated to sporulation plates overlaid with a nitrocellulose filter (Protran BA85, Whatman). After 3-day incubation at 30˚C, fluorescence was visualized by illuminating the open plates from the top with a hand-held 302-nm ultraviolet (UV) lamp. Images were taken using a Gel Doc XR system (Bio-Rad). Sporulation efficiency was quantitated by microscopic examination of asci formation after 3 days on sporulation plates. Both mature and immature asci were scored. At least 300 cells were counted for every strain. Spore viability was assessed by tetrad dissection. At least 216 spores were scored for every strain.

## Statistics

To determine the statistical significance of differences, a two-tailed Student t-test was used. *P*-values were calculated with the GraphPad Prism 8.0 software. $P<0.05$ (*); $P<0.01$ (**); $P<0.001$ (***); $P<0.0001$ (****). The nature of the error bars in the graphical representations and the number of biological replicates are indicated in the corresponding figure legend.

## Supporting information

**S1 Fig. Hop1 localization to axes is normal in *HOP1/HOP1-mCherry* strains.** Immunofluorescence of meiotic chromosomes stained with anti-Hop1 (red) and DAPI (blue). Arrows point to the rDNA region devoid of Hop1. Spreads were prepared at 16 h. Scale bar, 2 μm. Strains are: DP422 (*zip1Δ HOP1/HOP1*) and DP1500 (*zip1Δ HOP1/HOP1-mCherry*).
(TIF)

**S2 Fig. Redirecting Pch2 subcellular distribution by addition of NES or NLS sequences.**
**(A)** Fluorescence microscopy analysis of plasmid-expressed GFP-Pch2, GFP-NES-Pch2 or GFP-NLS-Pch2 (green) and Hop1-mCherry (red) in whole meiotic cells 15 h after meiotic induction. Representative cells are shown. Scale bar, 2 μm. **(B)** Quantification of the ratio of nuclear (including nucleolar) to cytoplasmic GFP fluorescent signal. Error bars: SD. The cartoon illustrates the subcellular localization of the different versions of GFP-Pch2 (green). The strain in (A) and (B) is DP1500 (*zip1Δ*) transformed with the centromeric plasmids pSS393 (*GFP-PCH2*), pSS408 *(GFP-NES-PCH2)* and pSS421 (*GFP-NLS-PCH2*).
(TIF)

**S3 Fig. Subcellular localization of GFP-NES-Pch2 and GFP-NLS-Pch2.** Additional representative fields corresponding to the fluorescence microscopy analysis of localization of GFP-Pch2, GFP-NES-Pch2 or GFP-NLS-Pch2 (green) and Hop1-mCherry (red) presented in Fig 3. Scale bar, 2 μm.
(TIF)

**S4 Fig. GFP-Pch2 localizes to foci on synapsed chromosomes alternating with Hop1.** Immunofluorescence of meiotic chromosomes stained with anti-GFP antibodies (to detect GFP-Pch2; green), anti-Hop1 antibodies (red) and DAPI (blue). White arrowhead points to the rDNA. Yellow arrows point to interstitial GFP-Pch2 foci alternating with Hop1 signal. The Pch2 signal was computer-enhanced to visualize chromosomal foci. Spreads were prepared from *ndt80Δ* strains at 24 h. Scale bar, 2 μm. The strain is DP1654.
(TIF)

**S5 Fig. Nup2 regulates Pch2 subcellular localization, but it is not required for the checkpoint triggered by *zip1Δ*. (A)** Fluorescence microscopy analysis of GFP-Pch2 distribution in whole meiotic cells of the indicated genotypes 16 hours after meiotic induction. Representative cells are shown. Arrows point to chromosomal (non-nucleolar) Pch2. Scale bar, 2 μm. **(B, C)** Quantification of the ratio of nuclear (including nucleolar) to cytoplasmic mean GFP fluorescent signal **(B)** and the nucleolar GFP-Pch2 signal **(C)** in cells analyzed as in **(A).** Error bars: SD. **(D)** Sporulation efficiency, assessed by microscopic counting of asci, and dityrosine fluorescence (DiTyr), as a visual indicator of sporulation, were examined after 3 days on sporulation plates. Error bars, SD; n = 3. At least 300 cells were counted for each strain. Strains in (A-C) are: DP1624 (*GFP-PCH2*), DP1744 (*nup2Δ GFP-PCH2*), DP1625 (*zip1Δ GFP-PCH2*) and DP1745 (*zip1Δ nup2Δ GFP-PCH2*). Strains in (D) are: DP1151 (wild type), DP1164 (*pch2Δ*), DP1723 (*nup2Δ*), DP1152 (*zip1Δ*), DP1161(*zip1Δ pch2Δ*) and DP1724 (*zip1Δ*

*nup2Δ*).
(TIF)

**S6 Fig. Pil1-GBP traps GFP-Pch2 at the plasma membrane triggering Mek1-dependent sporulation arrest.** **(A)** Fluorescence microscopy analysis of GFP-Pch2 and Pil1-GBP-mCherry distribution in whole meiotic cells of the indicated genotypes 16 hours after meiotic induction. Scale bar, 2 μm. **(B)** Immunofluorescence of meiotic chromosomes stained with anti-Hop1 (red) and anti-Nsr1 (nucleolar marker; green) antibodies, and DAPI (blue). The arrow points to the rDNA region. Spreads were prepared at 16 h. Scale bar, 2 μm. **(C)** Sporulation efficiency and dityrosine fluorescence (DiTyr), were examined after 3 days on sporulation plates. Error bars, SD; n = 3. At least 300 cells were counted for each strain. Strains in (A-C) are: DP1624 (*GFP-PCH2*), DP1797 (*GFP-PCH2 PIL1-GBP-mCherry*) and DP1813 (*GFP-PCH2 PIL1-GBP-mCherry mek1Δ*). **(D)** Deletion of *SPO11* alleviates the sporulation block resulting from GFP-NES-Pch2 tethering to the plasma membrane. Sporulation efficiency and dityrosine fluorescence (DiTyr), were analyzed as in (C). Error bars, SD; n = 3. Strains in (D) are: DP1523 (*spo11Δ*), DP1795 (*GFP-NES-PCH2 PIL1-GBP-mCherry*), DP1846 (*GFP-NES-PCH2 PIL1-GBP-mCherry spo11Δ*), DP1796 (*zip1Δ GFP-NES-PCH2 PIL1-GBP-mCherry*) and DP1847 (*zip1Δ GFP-NES-PCH2 PIL1-GBP-mCherry spo11Δ*).
(TIF)

**S7 Fig. Model of Pch2 action in different mutant conditions.** **(A)** In *zip1Δ pch2Δ*, the conformational change required for Hop1 chromosomal incorporation is inefficient leading to low levels of Hop1-T318 phosphorylation and checkpoint defects. **(B)** In *zip1Δ orc1-3mAID*, Pch2 is not recruited to the rDNA resulting in its accumulation in the cytoplasm fostering proficient Hop1 loading and activation. **(C)** In *zip1Δ NES-PCH2*, the balance of Pch2 distribution is biased to the cytoplasm also supporting checkpoint activation. **(D)** In *zip1Δ NLS-PCH2*, the balance of Pch2 distribution is skewed towards the nucleus resulting in the accumulation of the protein in the rDNA and the nucleoplasm. The checkpoint defect in *zip1Δ NLS-PCH2* likely stems from the reduced levels of cytoplasmic Pch2. However, since increased dosage of NLS-Pch2 causes a stronger checkpoint defect, it is possible that the accumulation of NLS-Pch2 in the nucleoplasm also exerts an inhibitory effect on checkpoint activity. **(E)** In *zip1Δ dot1Δ*, Pch2 loses its rDNA confinement and it is widely distributed throughout unsynapsed chromosomes provoking Hop1 release and, therefore, low levels of Hop1-T318 phosphorylation. **(F)** In *zip1Δ orc1-3mAID dot1Δ*, the inability of Pch2 to be recruited to the rDNA results in its exclusive cytoplasmic localization supporting checkpoint activation. Since in the absence of Orc1 there is no Pch2 to be confined in the rDNA, Dot1 is irrelevant in this context. **(G)** In *zip1Δ nup2Δ*, the pool of cytoplasmic Pch2 is increased likely reflecting a defect in Pch2 import to the nucleus in the absence of the nucleoporin; consequently, the amount of nucleolar Pch2 is reduced. The presence of Pch2 in the cytoplasm ensures an efficient checkpoint response. **(H)** In *zip1Δ PIL1-GBP*, the GFP-tagged Pch2 is sequestered in the eisosomes facing the cytoplasmic side of the plasma membrane and, therefore, being proficient in the generation of the Hop1 conformational state that facilitates chromosome incorporation. Furthermore, since in this situation Pch2 is tightly trapped outside the nucleus, any transient inhibitory effect of nuclear Pch2 is absent resulting in checkpoint hyperactivation.
(TIF)

**S8 Fig. Quantification of GFP-Pch2 distribution using total fluorescence values.** Quantification of the ratio of nuclear (including nucleolar) to cytoplasmic GFP fluorescent signal using total intensity values (integrated density) within the contour of the nuclear and cytoplasmic area selected. Error bars: SD. **(A)** Measurements corresponding to the experiments presented

in Fig 1C and 1D. **(B)** Measurements corresponding to the experiments presented in Fig 3A and 3B. **(C)** Measurements corresponding to the experiments presented in Fig 6A and 6B. **(D)** Measurements corresponding to the experiments presented in S5A and S5B Fig.
(TIF)

**S1 Table. *Saccharomyces cerevisiae* strains.**
(PDF)

**S2 Table. Plasmids.**
(PDF)

**S3 Table. Primary antibodies.**
(PDF)

**S1 File. Raw data.** Excel workbook with separate spreadsheets containing numerical data underlying the corresponding figure panels.
(XLSX)

**S2 File. Statistics summary.**
(XLSX)

## Acknowledgments

We are grateful to Andrés Clemente, David Álvarez-Melo and Félix Prado for helpful comments and discussions. We also thank Alfonso Fernández-Álvarez, Andrés Clemente and Scott Keeney for reagents, and Sara González-Arranz and Carlos R. Vázquez for advice on microscopy analysis.

## Author Contributions

**Conceptualization:** Esther Herruzo, Jesús A. Carballo, Pedro A. San-Segundo.

**Formal analysis:** Esther Herruzo, Ana Lago-Maciel.

**Funding acquisition:** Jesús A. Carballo, Pedro A. San-Segundo.

**Investigation:** Esther Herruzo, Ana Lago-Maciel, Sara Baztán, Pedro A. San-Segundo.

**Project administration:** Beatriz Santos.

**Resources:** Jesús A. Carballo.

**Supervision:** Beatriz Santos, Pedro A. San-Segundo.

**Visualization:** Esther Herruzo, Pedro A. San-Segundo.

**Writing – original draft:** Pedro A. San-Segundo.

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
