## [Decision Letter · Decision Letter 0]

21 May 2021

Dear Pedro,

Thank you very much for submitting your Research Article entitled 'Pch2 orchestrates the meiotic recombination checkpoint from the cytoplasm' to PLOS Genetics.

The manuscript was fully evaluated at the editorial level and by three independent peer reviewers. The reviewers appreciated the attention to an important topic but identified some concerns that we ask you address in a revised manuscript. I (ML) also have some comments, which are listed below.

We therefore ask you to modify the manuscript according to the review recommendations. Your revisions should address the specific points made by each reviewer and by myself.

3) Include with the next manuscript version numerical data supporting each graph and summary statistic in the figures. This is required by PLOS data availability policy (see below in this email) and is a condition for further consideration of the manuscript--it cannot wait until acceptance. My usual recommendation to authors is to include a supplementary file as an Excel workbook with a separate worksheet for each figure or figure panel, but other solutions are possible. Relevant panels are Fig 1a,d; Fig 2b,c; Fig3c,d; Fig4b,c; Fig5a,b,d; Fig6b,c,d; Fig7d,e; FigS2b, FigS4b,c,d; FigS5c--I hope that I have captured all of them.

My (ML's) comments:

1. The manuscript refers to heterozygous and homozygous pHOP1-GFP-PCH2 constructs, but it appears that most of the "heterozygous" constructs are in fact hemizygous (i.e. over a pch2∆). This distinction is important because a true heterozygous construct will express both GFP-Pch2 and wild type Pch2, which I think is not what you mean. Please revise the nomenclature to reflect this; alternatively, please consider as an alternate "1 copy" and "2 copies". Regardless of which nomenclature you settle on, please devote a paragraph, or at least a few sentences, at the beginning, to clearly explaining the situation, and why hemizygous constructs were mainly used (I assume that it is because the HOP1 promoter is stronger than the PCH2 promoter, and so a hemizygous GFP-PCH2 construct produces about the same amount of protein as a homozygous wild-type PCH2 strain).

2. If my reading of the materials and methods is correct, then the nuclear/cytoplasmic GFP ratio is comparing GFP signal/unit area, not total GFP signal in nucleus versus cytoplasm. This is fine, as long as the ratio of nucleus/cytoplasm total area is relatively invariant. But have you considered looking at total nucleus GFP/total cytoplasm GFP ratio? I think that it would reveal that there is more Pch2 is in the cytoplasm than in the nucleus in wild-type cells. Regardless of which ratio you use, it would be useful to say somewhere the average ratio of nucleus/cytoplasm total area, so an interested reader could make the calculation themselves.

3. lines 227-230. For the non-specialist reader, please say that Hop1-T318phos is being used as a measure of checkpoint activity.

4. Either in figures where t-test comparisons are made, or in the materials and methods, please indicate what p values an asterisk means (i.e. what is *,**,***, etc.)

5. Please quantify western blots (using Pgk1 as a standard). I think it's too much to include the numerical data in figures, so including the values in a supplementary file (the same one as is used for reporting other underlying numerical data would work) would be fine.

6. Reviewer #3 has suggested several additional experiments, all of which are nice ideas, but (as the reviewer says in their comments to the editor) none of which are in my opinion indispensable. I also think that, given the high levels of proteases in meiotic yeast cells, some of the suggested approaches (subcellular fractionation, co-IP) would be technically very challenging. 

7. Reviewer #2 commented on the references section, which I agree needs proofreading to fix all of the formatting inconsistencies that your favorite reference manager creates. I think that reviewer #2 meant that only proper names and the first word of the title should be *capitalized* (not italicized). Gene and species names, of course, should be italicized.

[LINK]

Please let us know if you have any questions while making these revisions. Thank you for submitting this very interesting paper for PLOS Genetics!

Yours sincerely,

Michael Lichten, Ph.D.

Associate Editor

PLOS Genetics

Gregory P. Copenhaver

Editor-in-Chief

PLOS Genetics

Reviewer's Responses to Questions

**Comments to the Authors:**

Reviewer #1: The manuscript “Pch2 orchestrates the meiotic recombination checkpoint from the cytoplasm” by Herruzo et al presents an exciting and timely study of how the Pch2 ATPase contributes to a critical checkpoint in meiotic cells. In S. cerevisiae meiosis, the chromosome axis protein Hop1 is required to recruit and control the activity of DNA-breakage and recombination complexes to promote recombination. At the same time, Hop1 is phosphorylated in response to unrepaired DNA breaks, which initiates a signaling cascade that eventually arrests cells and prevents them from completing meiosis. A critical protein in the checkpoint pathway is the AAA+ ATPase Pch2, but its roles have been poorly understood until recently. Biochemical and structural work has recently revealed that Pch2 converts HORMA domain proteins (of which Hop1 is one) from a peptide-bound “closed” conformation to an “unbuckled” state poised for binding another closure motif. Recent genetic work in S. cerevisiae (from the current group and others), plants, and animals has led to a developing overall model in which soluble Pch2 converts a self-closed, autoinhibited Hop1 to an unbuckled form capable of binding a closure motif in the axis protein Red1 to localize. Later in meiosis, Pch2 localizes to the growing synaptonemal complex and performs the same reaction on Hop1 to release it from chromosomes, deactivating the checkpoint and allowing meiotic progression. But, this model has not been so clearly stated and supported in earlier work as it is in this manuscript. The work uses creative localization determinants to show that soluble (cytoplasmic) Pch2 is responsible for activating the checkpoint, likely by opening Hop1 and thereby promoting its nuclear import and chromosome localization. Later, some nuclear Pch2 is required to inactivate the checkpoint and allow meiotic progression (as illustrated in the current plasma membrane localization experiments). Therefore I strongly endorse publication of this work, which is an important and timely synthesis of concepts that have been emerging but not yet clarified, after the authors address only a few minor questions and comments.

Questions/Comments:

The model of Pch2-mediated “unlocking” of Hop1 to mediate its nuclear import and Red1 association is compelling. However, Pch2 is not required for this process in an otherwise-normal meiosis: pch2-delta cells proceed through meiosis and have high spore viability, indicating that sufficient Hop1 and Red1 localize to chromosomes to support DSB formation, impose homolog bias through Hop1 phosphorylation and Mek1 activation, etc. Thus, Hop1 must have an inherent locked-unlocked equilibrium that enables limited recruitment to chromosomes without Pch2. I think it is important that the authors address this point alongside their discussion of Pch2’s role in situations where recombination or synapsis is compromised.

Can the authors briefly address why they used the HOP1 promoter to drive expression of their various PCH2 alleles?

I’m not sure the experiments in Figure S2 are necessary. These plasmid-based constructs are quickly (and correctly) substituted for GFP-fusion constructs integrated at the PCH2 locus. What does the inclusion of these strains/experiments add to the paper?

I would recommend moving Figure 3B to supplemental information or removing it entirely. To me it’s confusing to have essentially the same information presented in Figure 3A and 3B.

The panel labels in Figure 3A and 4A are somewhat confusing. I will leave it to the editor/journal to decide what kind of labeling changes to make.

For the model figure, I would recommend using “C-Hop1” and “U-Hop1” instead of “Hop1” and “Hop1*” as the two states. I understand the authors’ hesitance to make this strong prediction, but it is in fact what they propose in the Discussion, and it fits with all known biochemical and genetic data on the proteins in question. Then Hop1 would become C-Hop1 once again (bound to Red1 instead of its own C-terminus) when bound to chromosomes. Also, in the Mad2 literature, the term “Mad2*” was used historically to refer to one of the conformational states. I can’t find the exact references now, but it is probably best to avoid the asterisk in any case.

Just a note: After reading the passage on lines 654-663 summarizing the model, I raised my hands up and yelled “Yes!” - this is the best synthesis of the developing evidence and model that I have yet read.

Reviewer #2: The manuscript by Herruzo et al is a well written, thorough examination of where Pch2 functions in the cell to allow activation of the meiotic recombination checkpoint. Through an elegant set of experiments localizing Pch2 to either the nucleolus, chromosomes or the cytoplasm, the authors present compelling data that it is solely cytoplasmic Pch2 which is required for the checkpoint and that having too much nuclear Pch2 has deleterious consequences. Amazingly, they were able to tether Pch2 to eisosomes on the plasma membrane without affecting meiotic recombination checkpoint activity. This tethering was so complete that it resulted in constitutive checkpoint activation even in ZIP1 cells. The authors also show that Hop1 accumulation on chromosomes is promoted by cytoplasmic Pch2 and reduced when Pch2 is only in the nucleus. They propose a very interesting model that cytoplasmic Pch2 “unbuckles” Hop1 from its closure motif thereby allowing it to bind to unsynapsed chromosomes after it enters the nucleus. Phosphorylated Hop1 occurs in response to DSBs and is necessary for activation of the Mek1 effector kinase for the meiotic recombination checkpoint. They further propose that when Pch2 binds to synapsed chromosomes it removes Hop1, thereby reducing Mek1 kinase activity and abolishing the checkpoint. Additionally, the show that the role of DOT1 in the meiotic recombination checkpoint is to keep Pch2 in the nucleolus. Without DOT1, Pch2 can bind to chromosomes and remove Hop1, thereby preventing Mek1 activation. However, if Pch2 is artificially localized to the cytoplasm, there is no nuclear Pch2 to bind to chromosomes and therefore DOT1 is no longer relevant. Finally, they show that Nup2 promotes import of Pch2 into the nucleus.. The quality of the data is excellent and the experiments are well designed and interpreted. This work is a major step forward in our understanding of the molecular mechanisms by which meiotic recombination is regulated. I have only minor comments.

Line 89, It seems misleading to say that chromosome synapsis is occurring “in parallel” with recombination since recombination is required for SC formation.

Line 93, need a reference for the statement the SC provides an “environment for properly regulated recombination”.

Line 111 In zip1∆ (italicized), the...

Line 113, While it is true that Mek1 prevents Rad54-Rad51 complex formation, allowing these complexes to form is not sufficient to allow the high levels of IS repair observed in a mek1∆. Therefore Mek1 must have other targets that promote IH bias and this statement oversimplifies the situation. It would be more correct to say ...it inhibits DSB repair by intersister recombination in part by ....”

Line 128: The statement that Hop1 is excluded from fully synapsed chromosomes is commonly made but has always perplexed me given the many published pictures of SCs containing alternating domains of Hop1 and Zip1 by San-Segundo, Börner and other authors. So Hop1 is clearly on synapsed chromosomes. An explanation of this conundrum would be helpful.

Line 200, a reference should be provided for the HOP1 promoter. Out of curiosity, why didn’t the authors use the PCH2 promoter since it appears that the HOP1 promoter expresses PCH2 to a higher level than is observed with endogenous Pch2?

Line 259: the “Pch2 variants” are alleles of the gene so this should be italicized: PCH2 (italicized) variants

Line 269 genomically instead of genomic

Line 303, Consistent with...

Line 341 “...Hop1 axial binding..” is confusing, perhaps “...Hop1 binding to axial elements” or “...Hop1 binding to chromosome axes”.

Line 386: need a reference for using H3-T11 as a proxy for Mek1 activity

Line 407 and throughout: whenever the authors say the “absence of X”, the experiments that were done were genetic and therefore it is the absence of the gene, not the protein, that should be indicated.

Line 637: The authors make the intriguing suggestion both that the Hop1 closure motif is “unbuckled” from Hop1 in the cytoplasm and that this conformational change might expose a cryptic NLS. In support of this idea, the C-terminus of Hop1 where the closure motif is located is rich in arginine and lysines (RKISVSKKTLKSNW) which they might want to mention.

Line 686: Standard nomenclature is to indicate genes that have been replaced by a marker with “∆”, eg., zip1∆::LEU2. The presence of a colon alone indicates an insertion. Please add “∆” to all deletion alleles.

References: Only proper names and the first word of the title should be italicized. Also all gene names and genus species names should be italicized (et, ref 9, 10, 28 and throughout).

Reviewer #3: In this manuscript from the San-Segundo lab, the authors describe how the cytoplasmic fraction of Pch2 is capable to mediate the meiotic recombination checkpoint in yeast. Pch2 is a AAA+ ATPase that is essential to implement the meiotic arrest occurring in Zip1 mutants. During meiosis, Pch2 localizes at the nucleolus and along the chromosomes. Interestingly, in Zip1 mutants, Pch2 is only present in the nucleolus and this fraction of Pch2 is not required to implement the meiotic arrest. These observations lead the authors to hypothesize that another fraction of Pch2, presumably in the cytoplasm, could facilitate the meiotic checkpoint. To test this idea, authors forced Pch2 into the cytoplasm or the nucleus by fusing Nuclear Export Signal and Nuclear Localization Signal into the Pch2 gene. This strategy successfully localized Pch2 to the cytoplasm or the nucleus of the cells. Then the authors show how the strain with NES-Pch2 can properly implement the meiotic checkpoint, while the one with NLS-Pch2 is not. Furthermore, the accumulation of Pch2 in the nucleus has pathological consequences for the cell. These data make them suggest that Pch2 regulates the meiotic recombination checkpoint from the cytoplasm, proposing a model in which the cytoplasmic fraction of Pch2 causes a conformational change to Hop1, which prevents its incorporation into the chromosomes and its phosphorylation.

This study is well conducted, the rationale for each experiment is well explained, the results are reported in enough detail, and the conclusions are largely supported by the data. In my opinion, this study should be of interest to the wide readership of PLoS Genetics. I have some minor comments and questions, but generally, this is a clearly presented study with important ramifications for the field.

Comments:

-Please define NES and NLS in the abstract.

-To better characterize the efficiency of the strategy used to change the location of Pch2 within the different cell compartments, I would like the authors to show by Western Blot the presence of Pch2 in the cytoplasmic and the nuclear subcellular fractions, at least for the GFP-NES-PCH2 and the GFP-NLS-PCH2 strains.

- One prediction of the authors' proposed model would be that Hop1 and Pch2 would interact in the cytoplasm. I wonder if the authors could test it by co-IPing Pch2 or Hop1 from the cytoplasmic fraction of the different yeast strains used or by other methods like in situ proximity ligation assays.

-Please provide a WB showing the expression levels of the native Pch2 and the GFP-tagged Pch2 of homozygous and heterozygous strains containing GFP-NES-PCH2 and the GFP-NLS-PCH2.

-Fig. 2. Is the sporulation efficiency of Zip1� significantly different than the one from Zip1� GFP-NLS-PCH2 (heterozygous strain)? If not, please justify this result.

-line 277. Please provide evidence that suggests the nuclear fraction of GFP-NES-PCH2 is present in the nucleolus. I guess the authors infer this from the following experiments performed on spreads, but it is not clear from the current wording of this section.

-Fig. 3C. I think it would be good to provide the raw data for the GFP signal of the nucleus and the cytoplasm as a supplementary figure. It seems that Zip1 mutation causes a clear reduction in the ratio of nuclear/cytoplasmic GFP in all strains but GFP-NES-PCH2. Is this just because most of the GFP signal is already in the cytoplasm in wild-type strains?

- I think it would be good to show the background level of green fluorescence for the in vivo measurement experiments displayed in Fig. 3C, similar to what is displayed in 4B for pch2�.

- Line 503. Does the constitutive activation of the checkpoint observed in zip1� GFP-NES-PCH2 PIL1-GBP-mCh depends on SPO11?

**Have all data underlying the figures and results presented in the manuscript been provided?**

Reviewer #1: Yes

Reviewer #2: **No: **The authors say these data will be provided if the manuscript is accepted.

Reviewer #3: **No: **Authors should provide raw data for the quantification of the fluorescent signal, sporulation efficiency and meiosis progression.

PLOS authors have the option to publish the peer review history of their article (what does this mean?). If published, this will include your full peer review and any attached files.

Reviewer #1: No

Reviewer #2: No

Reviewer #3: No

---

## [Editor Report · Decision Letter 1]

25 Jun 2021

Dear Pedro,

We are pleased to inform you that your manuscript entitled "Pch2 orchestrates the meiotic recombination checkpoint from the cytoplasm" has been editorially accepted for publication in PLOS Genetics. Congratulations!

I (ML) have a few truly minor suggestions which you may wish to incorporate while preparing the final draft for the production team (the editorial team will not need to re-evaluate):

line 133: recombination in regions

line 172: fully active, strongly

line 485: Pil1-GBP-mCherry, displaying

line 613: also reveals the absence of Pch2 chromosomal signal

line 1176: suggest deleting "(asterisk)" or replacing with "(indicated by an asterisk)"

Yours sincerely,

Michael Lichten, Ph.D.

Associate Editor

PLOS Genetics

Gregory P. Copenhaver

Editor-in-Chief

PLOS Genetics

Comments from the reviewers (if applicable):

(Personal note from ML)--What a nice paper! I am very happy to have been able to work as AE on it.

**Data Deposition**

http://datadryad.org/submit?journalID=pgenetics&manu=PGENETICS-D-21-00557R1

**Press Queries**

---

## [Editor Report · Acceptance letter]

7 Jul 2021

PGENETICS-D-21-00557R1 

Pch2 orchestrates the meiotic recombination checkpoint from the cytoplasm 

Dear Dr San-Segundo, 

We are pleased to inform you that your manuscript entitled "Pch2 orchestrates the meiotic recombination checkpoint from the cytoplasm" has been formally accepted for publication in PLOS Genetics! Your manuscript is now with our production department and you will be notified of the publication date in due course.

With kind regards,

Katalin Szabo

PLOS Genetics

On behalf of:
